# Accelerated Stochastic Greedy Coordinate Descent by Soft Thresholding Projection onto Simplex

**Chaobing Song, Shaobo Cui, Yong Jiang, Shu-Tao Xia**
Tsinghua University
{songcb16,cuishaobo16}@mails.tsinghua.edu.cn
{jiangy, xiast}@sz.tsinghua.edu.cn *

## Abstract

In this paper we study the well-known greedy coordinate descent (GCD) algorithm to solve $\ell_1$-regularized problems and improve GCD by the two popular strategies: Nesterov's acceleration and stochastic optimization. Firstly, based on an $\ell_1$-norm square approximation, we propose a new rule for greedy selection which is non-trivial to solve but convex; then an efficient algorithm called "SOft ThreshOlding PrOjection (SOTOPO)" is proposed to exactly solve an $\ell_1$-regularized $\ell_1$-norm square approximation problem, which is induced by the new rule. Based on the new rule and the SOTOPO algorithm, the Nesterov's acceleration and stochastic optimization strategies are then successfully applied to the GCD algorithm. The resulted algorithm called accelerated stochastic greedy coordinate descent (ASGCD) has the optimal convergence rate $O(\sqrt{1/\epsilon})$; meanwhile, it reduces the iteration complexity of greedy selection up to a factor of sample size. Both theoretically and empirically, we show that ASGCD has better performance for high-dimensional and dense problems with sparse solutions.

## 1 Introduction

In large-scale convex optimization, first-order methods are widely used due to their cheap iteration cost. In order to improve the convergence rate and reduce the iteration cost further, two important strategies are used in first-order methods: Nesterov's acceleration and stochastic optimization. Nesterov's acceleration is referred to the technique that uses some algebra trick to accelerate first-order algorithms; while stochastic optimization is referred to the method that samples one training example or one dual coordinate at random from the training data in each iteration. Assume the objective function $F(x)$ is convex and smooth. Let $F^* = \min_{x \in R^d} F(x)$ be the optimal value. In order to find an approximate solution $x$ that satisfies $F(x) - F^* \leq \epsilon$, the vanilla gradient descent method needs $O(1/\epsilon)$ iterations. While after applying the Nesterov's acceleration scheme [16], the resulted accelerated full gradient method (AFG) [16] only needs $O(\sqrt{1/\epsilon})$ iterations, which is optimal for first-order algorithms [16]. Meanwhile, assume $F(x)$ is also a finite sum of $n$ sample convex functions. By sampling one training example, the resulted stochastic gradient descent (SGD) and its variants [13, 23, 1] can reduce the iteration complexity by a factor of the sample size. As an alternative of SGD, randomized coordinate descent (RCD) can also reduce the iteration complexity by a factor of the sample size [15] and obtain the optimal convergence rate $O(\sqrt{1/\epsilon})$ by Nesterov's acceleration [14, 12]. The development of gradient descent and RCD raises an interesting problem: can the Nesterov's acceleration and stochastic optimization strategies be used to improve other existing first-order algorithms?

In this paper, we answer this question partly by studying coordinate descent with Gauss-Southwell selection, i.e., greedy coordinate descent (GCD). GCD is widely used for solving sparse optimization problems in machine learning [22, 9, 17]. If an optimization problem has a sparse solution, it is more suitable than its counterpart RCD. However, the theoretical convergence rate is still $O(1/\epsilon)$. Meanwhile if the iteration complexity is comparable, GCD will be preferable than RCD [17]. However in the general case, in order to do exact Gauss-Southwell selection, computing the full gradient beforehand is necessary, which causes GCD has much higher iteration complexity than RCD. To be concrete, in this paper we consider the well-known nonsmooth $\ell_1$-regularized problem:

$$\min_{x \in \mathbb{R}^d} \left\{ F(x) \stackrel{\text{def}}{=} f(x) + \lambda \|x\|_1 \stackrel{\text{def}}{=} \frac{1}{n} \sum_{j=1}^{n} f_j(x) + \lambda \|x\|_1 \right\}, \tag{1}$$

where $\lambda \geq 0$ is a regularization parameter, $f(x) = \frac{1}{n} \sum_{j=1}^{n} f_j(x)$ is a smooth convex function that is a finite average of $n$ smooth convex function $f_j(x)$. Given samples $\{(a_1, b_1), (a_2, b_2), \ldots, (a_n, b_n)\}$ with $a_j \in \mathbb{R}^d$ $(j \in [n] \stackrel{\text{def}}{=} \{1, 2, \ldots, n\})$, if each $f_j(x) = f_j(a_j^T x, b_j)$, then (1) is an $\ell_1$-regularized empirical risk minimization ($\ell_1$-ERM) problem. For example, if $b_j \in \mathbb{R}$ and $f_j(x) = \frac{1}{2}(b_j - a_j^T x)^2$, (1) is Lasso; if $b_j \in \{-1, 1\}$ and $f_j(x) = \log(1 + \exp(-b_j a_j^T x))$, $\ell_1$-regularized logistic regression is obtained.

In the above nonsmooth case, the Gauss-Southwell rule has 3 different variants [17, 22]: GS-$s$, GS-$r$ and GS-$q$. The GCD algorithm with all the 3 rules can be viewed as the following procedure: in each iteration based on a quadratic approximation of $f(x)$ in (1), one minimizes a surrogate objective function under the constraint that the direction vector used for update has at most 1 nonzero entry. The resulted problems under the 3 rules are easy to solve but are *nonconvex* due to the cardinality constraint of direction vector. While when using Nesterov's acceleration scheme, convexity is needed for the derivation of the optimal convergence rate $O(\sqrt{1/\epsilon})$ [16]. Therefore, it is impossible to accelerate GCD by the Nesterov's acceleration scheme under the 3 existing rules.

In this paper, we propose a novel variant of Gauss-Southwell rule by using an $\ell_1$-norm square approximation of $f(x)$ rather than quadratic approximation. The new rule involves an $\ell_1$-*regularized* $\ell_1$-*norm square approximation* problem, which is nontrivial to solve but is *convex*. To *exactly* solve the challenging problem, we propose an efficient SOft ThreshOlding PrOjection (SOTOPO) algorithm. The SOTOPO algorithm has $O(d + |Q| \log |Q|)$ cost, where it is often the case $|Q| \ll d$. The complexity result $O(d + |Q| \log |Q|)$ is better than $O(d \log d)$ of its counterpart SOPOPO [18], which is an Euclidean projection method.

Then based on the new rule and SOTOPO, we accelerate GCD to attain the optimal convergence rate $O(\sqrt{1/\epsilon})$ by combing a delicately selected mirror descent step. Meanwhile, we show that it is not necessary to compute full gradient beforehand: sampling one training example and computing a noisy gradient rather than full gradient is enough to perform greedy selection. This stochastic optimization technique reduces the iteration complexity of greedy selection by a factor of the sample size. The final result is an accelerated stochastic greedy coordinate descent (ASGCD) algorithm.

Assume $x^*$ is an optimal solution of (1). Assume that each $f_j(x)$(for all $j \in [n]$) is $L_p$-smooth $w.r.t.$ $\| \cdot \|_p$ $(p = 1, 2)$, i.e., for all $x, y \in \mathbb{R}^d$,

$$\|\nabla f_j(x) - \nabla f_j(y)\|_q \leq L_p \|x - y\|_p, \tag{2}$$

where if $p = 1$, then $q = \infty$; if $p = 2$, then $q = 2$.

In order to find an $x$ that satisfies $F(x) - F(x^*) \leq \epsilon$, ASGCD needs $O\left( \frac{\sqrt{CL_1}\|x^*\|_1}{\sqrt{\epsilon}} \right)$ iterations (see (16)), where $C$ is a function of $d$ that varies slowly over $d$ and is upper bounded by $\log^2(d)$. For high-dimensional and dense problems with sparse solutions, ASGCD has better performance than the state of the art. Experiments demonstrate the theoretical result.

Notations: Let $[d]$ denote the set $\{1, 2, \ldots, d\}$. Let $\mathbb{R}_+$ denote the set of nonnegative real number. For $x \in \mathbb{R}^d$, let $\|x\|_p = (\sum_{i=1}^{d} |x_i|^p)^{\frac{1}{p}}$ $(1 \leq p < \infty)$ denote the $\ell_p$-norm and $\|x\|_\infty = \max_{i \in [d]} |x_i|$ denote the $\ell_\infty$-norm of $x$. For a vector $x$, let $\dim(x)$ denote the dimension of $x$; let $x_i$ denote the $i$-th element of $x$. For a gradient vector $\nabla f(x)$, let $\nabla_i f(x)$ denote the $i$-th element of $\nabla f(x)$. For a set $S$, let $|S|$ denote the cardinality of $S$. Denote the simplex $\triangle_d = \{\theta \in \mathbb{R}_+^d : \sum_{i=1}^{d} \theta_i = 1\}$.

## 2 The SOTOPO algorithm

The proposed SOTOPO algorithm aims to solve the proposed new rule, *i.e.*, minimize the following $\ell_1$-regularized $\ell_1$-norm square approximation problem,

$$\tilde{h} \stackrel{\text{def}}{=} \arg\min_{g \in \mathbb{R}^d} \left\{ \langle \nabla f(x), g \rangle + \frac{1}{2\eta} \|g\|_1^2 + \lambda \|x + g\|_1 \right\}, \tag{3}$$

$$\tilde{x} \stackrel{\text{def}}{=} x + \tilde{h}, \tag{4}$$

where $x$ denotes the current iteration, $\eta$ a step size, $g$ the variable to optimize, $\tilde{h}$ the director vector for update and $\tilde{x}$ the next iteration. The number of nonzero entries of $\tilde{h}$ denotes how many coordinates will be updated in this iteration. Unlike the quadratic approximation used in GS-$s$, GS-$r$ and GS-$q$ rules, in the new rule the coordinate(s) to update is implicitly selected by the sparsity-inducing property of the $\ell_1$-norm square $\|g\|_1^2$ rather than using the cardinality constraint $\|g\|_0 \leq 1$ (*i.e.*, $g$ has at most 1 nonzero element) [17, 22]. By [6, §9.4.2], when the nonsmooth term $\lambda\|x + g\|_1$ in (1) does not exist, the minimizer of the $\ell_1$-norm square approximation (*i.e.*, $\ell_1$-norm steepest descent) is equivalent to GCD. When $\lambda\|x + g\|_1$ exists, generally, there may be one or more coordinates to update in this new rule. Because of the sparsity-inducing property of $\|g\|_1^2$ and $\|x + g\|_1$, both the direction vector $\tilde{h}$ and the iterative solution $\tilde{x}$ are sparse. In addition, (3) is an unconstrained problem and thus is feasible.

### 2.1 A variational reformulation and its properties

(3) involves the nonseparable, nonsmooth term $\|g\|_1^2$ and the nonsmooth term $\|x + g\|_1$. Because there are two nonsmooth terms, it seems difficult to solve (3) directly. While by the variational identity $\|g\|_1^2 = \inf_{\theta \in \triangle_d} \sum_{i=1}^d \frac{g_i^2}{\theta_i}$ in [4] [2], in Lemma 1, it is shown that we can transform the original nonseparable and nonsmooth problem into a separable and smooth optimization problem on a simplex.

**Lemma 1.** *By defining*

$$J(g, \theta) \stackrel{\text{def}}{=} \langle \nabla f(x), g \rangle + \frac{1}{2\eta} \sum_{i=1}^d \frac{g_i^2}{\theta_i} + \lambda\|x + g\|_1, \tag{5}$$

$$\tilde{g}(\theta) \stackrel{\text{def}}{=} \arg\min_{g \in \mathbb{R}^d} J(g, \theta), \quad J(\theta) \stackrel{\text{def}}{=} J(\tilde{g}(\theta), \theta), \tag{6}$$

$$\tilde{\theta} \stackrel{\text{def}}{=} \arg\inf_{\theta \in \triangle_d} J(\theta), \tag{7}$$

*where $\tilde{g}(\theta)$ is a vector function. Then the minimization problem to find $\tilde{h}$ in (3) is equivalent to the problem (7) to find $\tilde{\theta}$ with the relation $\tilde{h} = \tilde{g}(\tilde{\theta})$. Meanwhile, $\tilde{g}(\theta)$ and $J(\theta)$ in (6) are both coordinate separable with the expressions*

$$\forall i \in [d], \ \tilde{g}_i(\theta) = \tilde{g}_i(\theta_i) \stackrel{\text{def}}{=} sign(x_i - \theta_i \eta \nabla_i f(x)) \cdot \max\{0, |x_i - \theta_i \eta \nabla_i f(x)| - \theta_i \eta \lambda\} - x_i, \tag{8}$$

$$J(\theta) = \sum_{i=1}^d J_i(\theta_i), \quad \text{where} \quad J_i(\theta_i) \stackrel{\text{def}}{=} \nabla_i f(x) \cdot \tilde{g}_i(\theta_i) + \frac{1}{2\eta} \sum_{i=1}^d \frac{\tilde{g}_i^2(\theta_i)}{\theta_i} + \lambda|x_i + \tilde{g}_i(\theta_i)|. \tag{9}$$

In Lemma 1, (8) is obtained by the iterative soft thresholding operator [5]. By Lemma 1, we can reformulate (3) into the problem (5), which is about two parameters $g$ and $\theta$. Then by the joint convexity, we swap the optimization order of $g$ and $\theta$. Fixing $\theta$ and optimizing with respect to (*w.r.t.*) $g$, we can get a closed form of $\tilde{g}(\theta)$, which is a vector function about $\theta$. Substituting $\tilde{g}(\theta)$ into $J(g, \theta)$, we get the problem (7) about $\theta$. Finally, the optimal solution $\tilde{h}$ in (3) can be obtained by $\tilde{h} = \tilde{g}(\tilde{\theta})$.

The explicit expression of each $J_i(\theta_i)$ can be given by substituting (8) into (9). Because $\theta \in \triangle_d$, we have for all $i \in [d], 0 \leq \theta_i \leq 1$. In the following Lemma 2, it is observed that the derivate $J_i'(\theta_i)$ can be a constant or have a piecewise structure, which is the key to deduce the SOTOPO algorithm.

**Lemma 2.** *Assume that for all $i \in [d]$, $J_i'(0)$ and $J_i'(1)$ have been computed. Denote $r_{i1} \overset{\text{def}}{=}$ $\frac{|x_i|}{\sqrt{-2\eta J_i'(0)}}$ and $r_{i2} \overset{\text{def}}{=} \frac{|x_i|}{\sqrt{-2\eta J_i'(1)}}$, then $J_i'(\theta_i)$ belongs to one of the 4 cases,*

$$(case\ a): J_i'(\theta_i) = 0, \quad 0 \leq \theta_i \leq 1, \qquad (case\ b): J_i'(\theta_i) = J_i'(0) < 0, \quad 0 \leq \theta_i \leq 1,$$

$$(case\ c): J_i'(\theta_i) = \begin{cases} J_i'(0), & 0 \leq \theta_i \leq r_{i1} \\ -\frac{x_i^2}{2\eta\theta_i^2}, & r_{i1} < \theta_i \leq 1 \end{cases}, \quad (case\ d): J_i'(\theta_i) = \begin{cases} J_i'(0), & 0 \leq \theta_i \leq r_{i1} \\ -\frac{x_i^2}{2\eta\theta_i^2}, & r_{i1} < \theta_i < r_{i2} \\ J_i'(1), & r_{i2} \leq \theta_i \leq 1 \end{cases}.$$

Although the formulation of $J_i'(\theta_i)$ is complicated, by summarizing the property of the 4 cases in Lemma 2, we have Corollary 1.

**Corollary 1.** *For all $i \in [d]$ and $0 \leq \theta_i \leq 1$, if the derivate $J_i'(\theta_i)$ is not always 0, then $J_i'(\theta_i)$ is a non-decreasing, continuous function with value always less than 0.*

Corollary 1 shows that except the trivial (case a), for all $i \in [d]$, whichever $J_i'(\theta_i)$ belong to (case b), (case c) or case (d), they all share the same group of properties, which makes a consistent iterative procedure possible for all the cases. The different formulations in the four cases mainly have impact about the stopping criteria of SOTOPO.

## 2.2 The property of the optimal solution

The Lagrangian of the problem (7) is

$$\mathcal{L}(\theta, \gamma, \zeta) \overset{\text{def}}{=} J(\theta) + \gamma \Big( \sum_{i=1}^{d} \theta_i - 1 \Big) - \langle \zeta, \theta \rangle, \tag{10}$$

where $\gamma \in \mathbb{R}$ is a Lagrange multiplier and $\zeta \in \mathbb{R}_+^d$ is a vector of non-negative Lagrange multipliers. Due to the coordinate separable property of $J(\theta)$ in (9), it follows that $\frac{\partial J(\theta)}{\partial \theta_i} = J_i'(\theta_i)$. Then the KKT condition of (10) can be written as

$$\forall i \in [d], \quad J_i'(\theta_i) + \gamma - \zeta_i = 0, \quad \zeta_i \theta_i = 0, \quad \text{and} \quad \sum_{i=1}^{d} \theta_i = 1. \tag{11}$$

By reformulating the KKT condition (11), we have Lemma 3.

**Lemma 3.** *If $(\tilde{\gamma}, \tilde{\theta}, \tilde{\zeta})$ is a stationary point of (10), then $\tilde{\theta}$ is an optimal solution of (7). Meanwhile, denote $S \overset{\text{def}}{=} \{i : \tilde{\theta}_i > 0\}$ and $T \overset{\text{def}}{=} \{j : \tilde{\theta}_j = 0\}$, then the KKT condition can be formulated as*

$$\begin{cases} \sum_{i \in S} \tilde{\theta}_i = 1; \\ for\ all\ j \in T, \quad \tilde{\theta}_j = 0; \\ for\ all\ i \in S, \quad \tilde{\gamma} = -J_i'(\tilde{\theta}_i) \geq \max_{j \in T} -J_j'(0). \end{cases} \tag{12}$$

By Lemma 3, if the set $S$ in Lemma 3 is known beforehand, then we can compute $\tilde{\theta}$ by simply applying the equations in (12). Therefore finding the optimal solution $\tilde{\theta}$ is equivalent to finding the set of the nonzero elements of $\tilde{\theta}$.

## 2.3 The soft thresholding projection algorithm

In Lemma 3, for each $i \in [d]$ with $\tilde{\theta}_i > 0$, it is shown that the negative derivate $-J_i'(\tilde{\theta}_i)$ is equal to a single variable $\tilde{\gamma}$. Therefore, a much simpler problem can be obtained if we know the coordinates of these positive elements. At first glance, it seems difficult to identify these coordinates, because the number of potential subsets of coordinates is clearly exponential on the dimension $d$. However, the property clarified by Lemma 2 enables an efficient procedure for identifying the nonzero elements of $\tilde{\theta}$. Lemma 4 is a key tool in deriving the procedure for identifying the non-zero elements of $\tilde{\theta}$.

**Lemma 4** (Nonzero element identification). *Let $\tilde{\theta}$ be an optimal solution of (7). Let $s$ and $t$ be two coordinates such that $J_s'(0) < J_t'(0)$. If $\tilde{\theta}_s = 0$, then $\tilde{\theta}_t$ must be 0 as well; equivalently, if $\tilde{\theta}_t > 0$, then $\tilde{\theta}_s$ must be greater than 0 as well.*

Lemma 4 shows that if we sort $u \stackrel{\text{def}}{=} -\nabla J(0)$ such that $u_{i_1} \geq u_{i_2} \geq \cdots \geq u_{i_d}$, where $\{i_1, i_2, \ldots, i_d\}$ is a permutation of $[d]$, then the set $S$ in Lemma 3 is of the form $\{i_1, i_2, \ldots, i_\varrho\}$, where $1 \leq \varrho \leq d$. If $\varrho$ is obtained, then we can use the fact that for all $j \in [\varrho]$,

$$-J'_{i_j}(\tilde{\theta}_{i_j}) = \tilde{\gamma} \quad \text{and} \quad \sum_{j=1}^{\varrho} \tilde{\theta}_{i_j} = 1 \tag{13}$$

to compute $\tilde{\gamma}$. Therefore, by Lemma 4, we can efficiently identify the nonzero elements of the optimal solution $\tilde{\theta}$ after a sort operation, which costs $O(d \log d)$. However based on Lemmas 2 and 3, the sort cost $O(d \log d)$ can be further reduced by the following Lemma 5.

**Lemma 5** (Efficient identification). *Assume $\tilde{\theta}$ and $S$ are given in Lemma 3. Then for all $i \in S$,*

$$-J'_i(0) \geq \max_{j \in [d]} \{-J'_j(1)\}. \tag{14}$$

By Lemma 5, before ordering $u$, we can filter out all the coordinates $i$'s that satisfy $-J'_i(0) < \max_{j \in [d]} -J'_j(1)$. Based on Lemmas 4 and 5, we propose the SOft ThreshOlding PrOjection (SOTOPO) algorithm in Alg. 1 to efficiently obtain an optimal solution $\tilde{\theta}$. In the step 1, by Lemma 5, we find the quantity $v_m, i_m$ and $Q$. In the step 2, by Lemma 4, we sort the elements $\{-J'_i(0)| \ i \in Q\}$. In the step 3, because $S$ in Lemma 3 is of the form $\{i_1, i_2, \ldots, i_\varrho\}$, we search the quantity $\rho$ from 1 to $|Q| + 1$ until a stopping criteria is met. In Alg. 1, the number of nonzero elements of $\tilde{\theta}$ is $\rho$ or $\rho - 1$. In the step 4, we compute the $\tilde{\gamma}$ in Lemma 3 according to the conditions. In the step 5, the optimal $\tilde{\theta}$ and the corresponding $\tilde{h}, \tilde{x}$ are given.

---

**Algorithm 1** $\tilde{x} = $SOTOPO$(\nabla f(x), x, \lambda, \eta)$

---

1. Find

$$(v_m, i_m) \stackrel{\text{def}}{=} (\max_{i \in [d]} \{-J'_i(1)\}, \ \arg\max_{i \in [d]} \{-J'_i(1)\}), Q \stackrel{\text{def}}{=} \{i \in [d]| -J'_i(0) > v_m\}.$$

2. Sort $\{-J'_i(0)| \ i \in Q\}$ such that $-J'_{i_1}(0) \geq -J'_{i_2}(0) \geq \cdots \geq -J'_{i_{|Q|}}(0)$, where $\{i_1, i_2, \ldots, i_{|Q|}\}$ is a permutation of the elements in $Q$. Denote

$$v \stackrel{\text{def}}{=} (-J'_{i_1}(0), -J'_{i_2}(0), \ldots, -J'_{i_{|Q|}}(0), v_m), \quad \text{and} \quad i_{|Q|+1} \stackrel{\text{def}}{=} i_m, \ v_{|Q|+1} \stackrel{\text{def}}{=} v_m.$$

3. For $j \in [|Q| + 1]$, denote $R_j = \{i_k | k \in [j]\}$. Search from 1 to $|Q| + 1$ to find the quantity

$$\rho \stackrel{\text{def}}{=} \min \Big\{ j \in [|Q| + 1]| \ J'_{i_j}(0) = J'_{i_j}(1) \ \text{ or } \ \sum_{l \in R_j} |x_l| \geq \sqrt{2\eta v_j} \ \text{ or } \ j = |Q| + 1 \Big\}.$$

4. The $\tilde{\gamma}$ in Lemma 3 is given by

$$\tilde{\gamma} = \begin{cases} \Big(\sum_{l \in R_{\rho-1}} |x_l|\Big)^2 / (2\eta), & \text{if } \sum_{l \in R_{\rho-1}} |x_l| \geq \sqrt{2\eta v_\rho}; \\ v_\rho, & \text{otherwise.} \end{cases}$$

5. Then the $\tilde{\theta}$ in Lemma 3 and its corresponding $\tilde{h}, \tilde{x}$ in (3) and (4) are obtained by

$$(\tilde{\theta}_l, \tilde{h}_l, \tilde{x}_l) = \begin{cases} \big(\frac{|x_l|}{\sqrt{2\eta\tilde{\gamma}}}, -x_l, 0\big), & \text{if } l \in R_\rho \backslash \{i_\rho\}; \\ \big(1 - \sum_{k \in R_\rho \backslash \{i_\rho\}} \tilde{\theta}_k, \ \tilde{g}_l(\tilde{\theta}_l), \ x_l + \tilde{g}_l(\tilde{\theta}_l)\big), & \text{if } l = i_\rho; \\ (0, 0, x_l), & \text{if } l \in [d] \backslash R_\rho. \end{cases}$$

---

In Theorem 1, we give the main result about the SOTOPO algorithm.

**Theorem 1.** *The SOTOPO algorihtm in Alg. 1 can get the exact minimizer $\tilde{h}, \tilde{x}$ of the $\ell_1$-regularized $\ell_1$-norm square approximation problem in (3) and (4).*

The SOTOPO algorithm seems complicated but is indeed efficient. The dominant operations in Alg. 1 are steps 1 and 2 with the total cost $O(d + |Q| \log |Q|)$. To show the effect of the complexity reduction by Lemma 5, we give the following fact.

**Proposition 1.** *For the optimization problem defined in* (5)-(7)*, where $\lambda$ is the regularization parameter of the original problem* (1)*, we have that*

$$0 \leq \max_{i\in[d]}\left\{\sqrt{\frac{-2J_i'(0)}{\eta}}\right\} - \max_{j\in[d]}\left\{\sqrt{\frac{-2J_j'(1)}{\eta}}\right\} \leq 2\lambda. \tag{15}$$

Assume $v_m$ is defined in the step 1 of Alg. 1. By Proposition 1, for all $i \in Q$,

$$\sqrt{\frac{-2J_i'(0)}{\eta}} \leq \max_{k\in[d]}\left\{\sqrt{\frac{-2J_k'(0)}{\eta}}\right\} \leq \max_{j\in[d]}\left\{\sqrt{\frac{-2J_j'(1)}{\eta}}\right\} + 2\lambda = \sqrt{\frac{2v_m}{\eta}} + 2\lambda,$$

Therefore at least the coordinates $j$'s that satisfy $\sqrt{\frac{-2J_j'(0)}{\eta}} > \sqrt{\frac{2v_m}{\eta}} + 2\lambda$ will be not contained in $Q$. In practice, it can considerably reduce the sort complexity.

**Remark 1.** *SOTOPO can be viewed as an extension of the SOPOPO algorithm [18] by changing the objective function from Euclidean distance to a more general function $J(\theta)$ in* (9)*. It should be noted that Lemma 5 does not have a counterpart in the case that the objective function is Euclidean distance [18]. In addition, some extension of randomized median finding algorithm [10] with linear time in our setting is also deserved to research. Due to the limited space, it is left for further discussion.*

## 3 The ASGCD algorithm

Now we can come back to our motivation, *i.e.*, accelerating GCD to obtain the optimal convergence rate $O(1/\sqrt{\epsilon})$ by Nesterov's acceleration and reducing the complexity of greedy selection by stochastic optimization. The main idea is that although like any (block) coordinate descent algorithm, the proposed new rule, *i.e.*, minimizing the problem in (3), performs update on one or several coordinates, it is a generalized proximal gradient descent problem based on $\ell_1$-norm. Therefore this rule can be applied into the existing Nesterov's acceleration and stochastic optimization framework "Katyusha" [1] if it can be solved efficiently. The final result is the accelerated stochastic greedy coordinate descent (ASGCD) algorithm, which is described in Alg. 2.

---

**Algorithm 2** ASGCD

---
$\delta = \log(d) - 1 - \sqrt{(\log(d)-1)^2 - 1}$;
$p = 1 + \delta, q = \frac{p}{p-1}, C = \frac{d^{\frac{2\delta}{1+\delta}}}{\delta}$;
$z_0 = y_0 = \tilde{x}_0 = \vartheta_0 = 0$;
$\tau_2 = \frac{1}{2}, m = \lceil\frac{n}{b}\rceil, \eta = \frac{1}{\left(1+2\frac{n-b}{b(n-1)}\right)L_1}$;
**for** $s = 0, 1, 2, \ldots, S-1$, do

    1. $\tau_{1,s} = \frac{2}{s+4}, \alpha_s = \frac{\eta}{\tau_{1,s}C}$;

    2. $\mu_s = \nabla f(\tilde{x}_s)$;

    3. **for** $l = 0, 1, \ldots, m-1$, do

        (a) $k = (sm) + l$;
        (b) randomly sample a mini batch $\mathcal{B}$ of size b from $\{1, 2, \ldots, n\}$ with equal probability;
        (c) $x_{k+1} = \tau_{1,s}z_k + \tau_2\tilde{x}_s + (1 - \tau_{1,s} - \tau_2)y_k$;
        (d) $\tilde{\nabla}_{k+1} = \mu_s + \frac{1}{b}\sum_{j\in\mathcal{B}}(\nabla f_j(x_{k+1}) - \nabla f_j(\tilde{x}_s))$;
        (e) $y_{k+1} = $SOTOPO$(\tilde{\nabla}_{k+1}, x_{k+1}, \lambda, \eta)$;
        (f) $(z_{k+1}, \vartheta_{k+1}) = $pCOMID$(\tilde{\nabla}_{k+1}, \vartheta_k, q, \lambda, \alpha_s)$;

    **end for**

    4. $\tilde{x}_{s+1} = \frac{1}{m}\sum_{l=1}^{m} y_{sm+l}$;

**end for**
**Output:** $\tilde{x}_S$

---

---

**Algorithm 3** $(\tilde{x}, \tilde{\vartheta}) = \text{pCOMID}(g, \vartheta, q, \lambda, \alpha)$

---

    1. $\forall i \in [d], \tilde{\vartheta}_i = \text{sign}(\vartheta_i - \alpha g_i) \cdot \max\{0, |\vartheta_i - \alpha g_i| - \alpha\lambda\};$

    2. $\forall i \in [d], \tilde{x}_i = \frac{\text{sign}(\tilde{\vartheta}_i)|\tilde{\theta}_i|^{q-1}}{\|\tilde{\vartheta}\|_q^{q-2}};$

    3. **Output:** $\tilde{x}, \tilde{\vartheta}$.

---

In Alg. 2, the gradient descent step $3(e)$ is solved by the proposed SOTOPO algorithm, while the mirror descent step $3(f)$ is solved by the COMID algorithm with $p$-norm divergence [11, Sec. 7.2]. We denote the mirror descent step as pCOMID in Alg. 3. All other parts are standard steps in the Katyusha framework except some parameter settings. For example, instead of the custom setting $p = 1 + 1/\log(d)$ [19, 11], a particular choice $p = 1 + \delta$ ($\delta$ is defined in Alg. 2) is used to minimize the $C = \frac{d^{\frac{2\delta}{1+\delta}}}{\delta}$. $C$ varies slowly over $d$ and is upper bounded by $\log^2(d)$. Meanwhile, $\alpha_{k+1}$ depends on the extra constant $C$. Furthermore, the step size $\eta = \frac{1}{\left(1 + 2\frac{n-b}{b(n-1)}\right)L_1}$ is used, where $L_1$ is defined in (2). Finally, unlike [1, Alg. 2], we let the batch size $b$ as an algorithm parameter to cover both the stochastic case $b < n$ and the deterministic case $b = n$. To the best of our knowledge, the existing GCD algorithms are deterministic, therefore by setting $b = n$, we can compare with the existing GCD algorithms better.

Based on the efficient SOTOPO algorithm, ASGCD has nearly the same iteration complexity with the standard form [1, Alg. 2] of Katyusha. Meanwhile we have the following convergence rate.

**Theorem 2.** *If each $f_j(x)(j \in [n])$ is convex, $L_1$-smooth in (2) and $x^*$ is an optimum of the $\ell_1$-regularized problem (1), then ASGCD satisfies*

$$\mathbb{E}[F(\tilde{x}^S)] - F(x^*) \leq \frac{4}{(S+3)^2}\left(1 + \frac{1 + 2\beta(b)}{2m}C\right)L_1\|x^*\|_1^2 = O\left(\frac{CL_1\|x^*\|_1^2}{S^2}\right), \quad (16)$$

*where $\beta(b) = \frac{n-b}{b(n-1)}$, $S$, $b$, $m$ and $C$ are given in Alg. 2. In other words, ASGCD achieves an $\epsilon$-additive error (i.e., $\mathbb{E}[F(\tilde{x}^S)] - F(x^*) \leq \epsilon$ ) using at most $O\left(\frac{\sqrt{CL_1}\|x^*\|_1}{\sqrt{\epsilon}}\right)$ iterations.*

In Table 1, we give the convergence rate of the existing algorithms and ASGCD to solve the $\ell_1$-regularized problem (1). In the first column, "Acc" and "Non-Acc" denote the corresponding algorithms are Nesterov's accelerated or not respectively, "Primal" and "Dual" denote the corresponding algorithms solves the primal problem (1) and its regularized dual problem [20] respectively, $\ell_2$-norm and $\ell_1$-norm denote the theoretical guarantee is based on $\ell_2$-norm and $\ell_1$-norm respectively. In terms of $\ell_2$-norm based guarantee, Katyusha and APPROX give the state of the art convergence rate $O\left(\frac{\sqrt{L_2}\|x^*\|_2}{\sqrt{\epsilon}}\right)$. In terms of $\ell_1$-norm based guarantee, GCD gives the state of the art convergence rate $O(\frac{L_1\|x\|_1^2}{\epsilon})$, which is only applicable for the smooth case $\lambda = 0$ in (1). When $\lambda > 0$, the generalized GS-$r$, GS-$s$ and GS-$q$ rules generally have worse theoretical guarantee than GCD [17]. While the bound of ASGCD in this paper is $O(\frac{\sqrt{L_1}\|x\|_1 \log d}{\sqrt{\epsilon}})$, which can be viewed as an accelerated version of the $\ell_1$-norm based guarantee $O(\frac{L_1\|x\|_1^2}{\epsilon})$. Meanwhile, because the bound depends on $\|x^*\|_1$ rather than $\|x^*\|_2$ and on $L_1$ rather than $L_2$ ($L_1$ and $L_2$ are defined in (2)), for the $\ell_1$-ERM problem, if the samples are high-dimensional, dense and the regularization parameter $\lambda$ is relatively large, then it is possible that $L_1 \ll L_2$ (in the extreme case, $L_2 = dL_1$ [9]) and $\|x^*\|_1 \approx \|x^*\|_2$. In this case, the $\ell_1$-norm based guarantee $O(\frac{\sqrt{L_1}\|x\|_1 \log d}{\sqrt{\epsilon}})$ of ASGCD is better than the $\ell_2$-norm based guarantee $O\left(\frac{\sqrt{L_2}\|x^*\|_2}{\sqrt{\epsilon}}\right)$ of Katyusha and APPROX. Finally, whether the $\log d$ factor in the bound of ASGCD (which also appears in the COMID [11] analysis) is necessary deserves further research.

**Remark 2.** *When the batch size $b = n$, ASGCD is a deterministic algorithm. In this case, we can use a better smooth constant $T_1$ that satisfies $\|\nabla f(x) - \nabla f(y)\|_\infty \leq T_1\|x - y\|_1$ rather than $L_1$ [1].*

**Remark 3.** *The necessity of computing the full gradient beforehand is the main bottleneck of GCD in applications [17]. There exists some work [9] to avoid the computation of full gradient by performing some approximate greedy selection. While the method in [9] needs preprocessing, incoherence*

Table 1: Convergence rate on $\ell_1$-regularized empirical risk minimization problems. (For GCD, the convergence rate is applied for $\lambda = 0$. )

| Algorithm Type | Paper | Convergence Rate |
|---|---|---|
| Non-Acc, Primal, $\ell_2$-norm | SAGA [8] | $O\left(\frac{L_2\|x^*\|_2^2}{\epsilon}\right)$ |
| Acc, Primal, $\ell_2$-norm | Katyusha [1] | $O\left(\frac{\sqrt{L_2}\|x^*\|_2}{\sqrt{\epsilon}}\right)$ |
| Acc, Dual, $\ell_2$-norm | Acc-SDCA [21] SPDC [24] APCG [14] APPROX [12] | $O\left(\frac{\sqrt{L_2}\|x^*\|_2}{\sqrt{\epsilon}} \log(\frac{1}{\epsilon})\right)$ |
| Non-Acc, Primal, $\ell_1$-norm | GCD [2] | $O\left(\frac{L_1\|x^*\|_1^2}{\epsilon}\right)$ |
| Acc, Primal, $\ell_1$-norm | ASGCD (**This Paper**) | $O\left(\frac{\sqrt{L_1}\|x^*\|_1 \log d}{\sqrt{\epsilon}}\right)$ |

*condition for dataset and is somewhat complicated. Contrary to [9], the proposed ASGCD algorithm reduces the complexity of greedy selection by a factor up to $n$ in terms of the amortized cost by simply applying the existing stochastic variance reduction framework.*

## 4 Experiments

In this section, we use numerical experiments to demonstrate the theoretical results in Section 3 and show the empirical performance of ASGCD with batch size $b = 1$ and its deterministic version with $b = n$ (In Fig. 1 they are denoted as ASGCD ($b = 1$) and ASGCD ($b = n$) respectively). In addition, following the claim to using data access rather than CPU time [19] and the recent SGD and RCD literature [13, 14, 1], we use the data access, i.e., the number of times the algorithm accesses the data matrix, to measure the algorithm performance. To show the effect of Nesterov's acceleration, we compare ASGCD ($b = n$) with the non-accelerated greedy coordinate descent with GS-$q$ rule, i.e., coordinate gradient descent (CGD) [22]. To show the effect of both Nesterov's acceleration and stochastic optimization strategies, we compare ASGCD ($b = 1$) with Katyusha [1, Alg. 2]. To show the effect of the proposed new rule in Section 2, which is based on $\ell_1$-norm square approximation, we compare ASGCD ($b = n$) with the $\ell_2$-norm based proximal accelerated full gradient (AFG) implemented by the linear coupling framework [3]. Meanwhile, as a benchmark of stochastic optimization for the problems with finite-sum structure, we also show the performance of proximal stochastic variance reduced gradient (SVRG) [23]. In addition, based on [1] and our experiments, we find that "Katyusha" [1, Alg. 2] has the best empirical performance in general for the $\ell_1$-regularized problem (1). Therefore other well-known state-of-art algorithms, such as APCG [14] and accelerated SDCA [21], are not included in the experiments.

The datasets are obtained from LIBSVM data [7] and summarized in Table 2. All the algorithms are used to solve the following lasso problem

$$\min_{x \in \mathbb{R}^d} \{f(x) + \lambda\|x\|_1 = \frac{1}{2n}\|b - Ax\|_2^2 + \lambda\|x\|_1\} \tag{17}$$

on the 3 datasets, where $A = (a_1, a_2, \ldots, a_n)^T = (h_1, h_2, \ldots, h_d) \in \mathbb{R}^{n \times d}$ with each $a_j \in \mathbb{R}^d$ representing a sample vector and $h_i \in \mathbb{R}^n$ representing a feature vector, $b \in \mathbb{R}^n$ is the prediction vector.

Table 2: Characteristics of three real datasets.

| Dataset name | # samples $n$ | # features $d$ |
|---|---|---|
| Leukemia | 38 | 7129 |
| Gisette | 6000 | 5000 |
| Mnist | 60000 | 780 |

For ASGCD ($b = 1$) and Katyusha [1, Alg. 2], we can use the tight smooth constant $L_1 = \max_{j \in [n], i \in [d]} |a_{j,i}^2|$ and $L_2 = \max_{j \in [n]} \|a_j\|_2^2$ respectively in their implementation. While for AS-

| $\lambda$ | Leu | Gisette | Mnist |
|---|---|---|---|

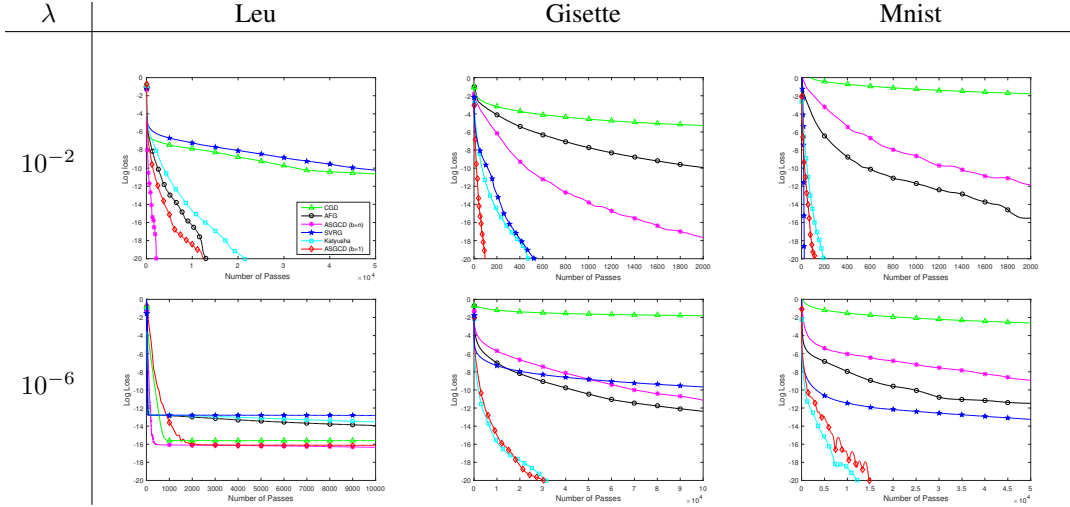

Figure 1: Comparing AGCD ($b = 1$) and ASGCD ($b = n$) with CGD, SVRG, AFG and Katyusha on Lasso.

GCD ($b = n$) and AFG, the better smooth constant $T_1 = \frac{\max_{i \in [d]} \|h_i\|_2^2}{n}$ and $T_2 = \frac{\|A\|^2}{n}$ are used respectively. The learning rate of CGD and SVRG are tuned in $\{10^{-6}, 10^{-5}, 10^{-4}, 10^{-3}, 10^{-2}, 10^{-1}\}$.

Table 3: Factor rates of for the 6 cases

| $\lambda$ | LEU | GISETTE | MNIST |
|---|---|---|---|
| $10^{-2}$ | $(0.85, 1.33)$ | $(0.88, 0.74)$ | $(5.85, 3.02)$ |
| $10^{-6}$ | $(1.45, 2.27)$ | $(3.51, 2.94)$ | $(5.84, 3.02)$ |

We use $\lambda = 10^{-6}$ and $\lambda = 10^{-2}$ in the experiments. In addition, for each case (Dataset, $\lambda$), AFG is used to find an optimum $x^*$ with enough accuracy.

The performance of the 6 algorithms is plotted in Fig. 1. We use Log loss $\log(F(x_k) - F(x^*))$ in the $y$-axis. $x$-axis denotes the number that the algorithm access the data matrix $A$. For example, ASGCD ($b = n$) accesses $A$ once in each iteration, while ASGCD ($b = 1$) accesses $A$ twice in an entire outer iteration. For each case (Dataset, $\lambda$), we compute the rate $(r_1, r_2) = \left( \frac{\sqrt{CL_1}\|x^*\|_1}{\sqrt{L_2}\|x^*\|_2}, \frac{\sqrt{CT_1}\|x^*\|_1}{\sqrt{T_2}\|x^*\|_2} \right)$ in Table 3. First, because of the acceleration effect, ASGCD ($b = n$) are always better than the non-accelerated CGD algorithm; second, by comparing ASGCD($b = 1$) with Katyusha and ASGCD ($b = n$) with AFG, we find that for the cases (Leu, $10^{-2}$), (Leu, $10^{-6}$) and (Gisette, $10^{-2}$), ASGCD ($b = 1$) dominates Katyusha [1, Alg.2] and ASGCD ($b = n$) dominates AFG. While the theoretical analysis in Section 3 shows that if $r_1$ is relatively small such as around 1, then ASGCD ($b = 1$) will be better than [1, Alg.2]. For the other 3 cases, [1, Alg.2] and AFG are better. The consistency between Table 3 and Fig. 1 demonstrates the theoretical analysis.

## Footnotes

*This work is supported by the National Natural Science Foundation of China under grant Nos. 61771273, 61371078.

[2] The infima can be replaced by minimization if the convention "$0/0 = 0$" is used.

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
