[Supplementary Material]

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

 [5] and the definition of $J(g, \theta)$ in (3), it follows that (3) can be rewritten as

$$\tilde{h} = \arg\min_{g \in \mathbb{R}^d} \{ \inf_{\theta \in \triangle_d} J(g, \theta) \}.$$

By the joint convexity of $J(g, \theta)$, we can find the minimizer $\tilde{h}$ by swapping the optimization order of $g$ and $\theta$, which is to say based on the definition of $\tilde{g}(\theta)$, $J(\theta)$ and $\tilde{\theta}$, we have

$$\tilde{h} = \tilde{g}(\tilde{\theta}).$$

Therefore, the minimization problem to find $\tilde{h}$ in (3) can be equivalently transformed to the problem (7). Meanwhile, it is observed that $J(g, \theta)$ is coordinate separable, *i.e.*,

$$J(g, \theta) = \sum_{i=1}^d J_i(g_i, \theta_i), \quad \text{where } J_i(g_i, \theta_i) \stackrel{\text{def}}{=} \nabla_i f(x) g_i + \frac{1}{2\eta} \frac{g_i^2}{\theta_i} + \lambda |x_i + g_i|. \tag{A.1}$$

By the definition of $\tilde{g}(\theta)$ in (6), $\tilde{g}(\theta)$ is also coordinate separable, *i.e.* for all $i \in [d]$,

$$\tilde{g}_i(\theta) = \tilde{g}_i(\theta_i) \stackrel{\text{def}}{=} \arg\min_{g_i \in \mathbb{R}} \left\{ \nabla_i f(x) g_i + \frac{1}{2\eta} \frac{g_i^2}{\theta_i} + \lambda |x_i + g_i| \right\}.$$

By using the iterative soft thresholding (IST) operator [7], for all $i \in [d]$,

$$\tilde{g}_i(\theta_i) = \text{sign}(x_i - \theta_i \eta \nabla_i f(x)) \cdot \max\{0, |x_i - \theta_i \eta \nabla_i f(x)| - \theta_i \eta \lambda\} - x_i. \tag{A.2}$$

Then it implies that $J(\theta)$ is also coordinate separable, *i.e.*,

$$J(\theta) = \sum_{i=1}^d J_i(\theta_i), \quad \text{where} \quad J_i(\theta_i) \stackrel{\text{def}}{=} J_i(\tilde{g}_i(\theta_i), \theta_i). \tag{A.3}$$

$\square$

## B Proofs of Lemma 2, Corollary 1 and Proposition 1

*Proof of Lemma 2.* For all $i \in [d]$, due to $\theta \in \triangle_d$, we have $0 \le \theta_i \le 1$. By substituting (8) into (9), we get the expression of $J_i(\theta_i)$. Taking the derivate of $J_i(\theta_i)$ and setting $\theta_i = 0, 1$ respectively, then we get the expressions of $J_i'(0), J_i'(1)$ as follows.

For all $i \in [d]$ and $\theta_i \ge 0$, denote

$$\nu_i \stackrel{\text{def}}{=} -\frac{(\max\{|\nabla_i f(x)| - \lambda, 0\})^2 \eta}{2}, \qquad \chi_i(\theta_i) \stackrel{\text{def}}{=} -\frac{(\text{sign}(x_i - \theta_i \eta \nabla_i f(x))\lambda + \nabla_i f(x))^2 \eta}{2}, \tag{B.1}$$

then the derivate $J_i'(\theta_i)$ at $\theta_i = 0, 1$ are

$$J_i'(0) = \begin{cases} \nu_i, & x_i = 0, \\ \chi_i(0), & x_i \ne 0 \end{cases}, \qquad J_i'(1) = \begin{cases} -\frac{x_i^2}{2\eta}, & |x_i - \eta \nabla_i f(x)| \le \eta \lambda \\ \chi_i(1), & |x_i - \eta \nabla_i f(x)| > \eta \lambda \end{cases}. \tag{B.2}$$

For all $i \in [d]$, according to the values of $x_i$ and $\nabla_i f(x)$, by classified discussion, we can show that $J_i'(\theta_i)$ belongs to one of the 4 cases in Lemma 2. Assume that $r_{i1}$ and $r_{i2}$ have been defined in

Lemma 2. Firstly, we denote

$$O \overset{\text{def}}{=} \{i | 0 \in \nabla_i f(x) + \lambda \partial |x_i| \}, \tag{B.3}$$

$$\begin{aligned}
U \overset{\text{def}}{=} \ & \{i \in [d] | x_i \geq 0, \nabla_i f(x) < -\lambda \} \\
& \cup \{i \in [d] | x_i \leq 0, \nabla_i f(x) > \lambda \} \\
& \cup \{i \in [d] | x_i > 0, \nabla_i f(x) > -\lambda, r_{i1} \geq 1 \} \\
& \cup \{i \in [d] | x_i < 0, \nabla_i f(x) < \lambda, r_{i1} \geq 1 \},
\end{aligned} \tag{B.4}$$

$$\begin{aligned}
V \overset{\text{def}}{=} \ & \{i \in [d] | x_i > 0, -\lambda < \nabla_i f(x) \leq \lambda, r_{i1} < 1 \} \\
& \cup \{i \in [d] | x_i < 0, -\lambda \leq \nabla_i f(x) < \lambda, r_{i1} < 1 \} \\
& \cup \{i \in [d] | x_i > 0, \nabla_i f(x) > \lambda, r_{i2} \geq 1 \} \\
& \cup \{i \in [d] | x_i < 0, \nabla_i f(x) < -\lambda, r_{i2} \geq 1 \},
\end{aligned} \tag{B.5}$$

$$\begin{aligned}
W \overset{\text{def}}{=} \ & \{i \in [d] | x_i > 0, \nabla_i f(x) > \lambda, r_{i2} < 1 \} \\
& \cup \{i \in [d] | x_i < 0, \nabla_i f(x) < -\lambda, r_{i2} < 1 \}.
\end{aligned} \tag{B.6}$$

Then based on the expressions of $J_i(\theta_i)$ in (A.1), (A.1) and (A.3), we can summarize the results as follows

- If $i \in O$, then $J_i'(\theta_i)$ belongs to the (case a) in Lemma 2.

- If $i \in U$, then $J_i'(\theta_i)$ belongs to the (case b) in Lemma 2.

- If $i \in V$, then $J_i'(\theta_i)$ belongs to the (case c) in Lemma 2.

- If $i \in W$, then $J_i'(\theta_i)$ belongs to the (case d) in Lemma 2.

$\square$

*Proof of Corollary 1.* Corollary 1 can be obtained by simply summarizing the 4 cases in Lemma 2. $\square$

*Proof of Proposition 1.* Assume that $\chi_i(\theta_i)$ is defined in (B.1), $O, U, V$ and $W$ are defined in (B.3)-(B.6).

Firstly, by checking $i \in O, U, V$ or $W$ orderly and using the expression of $J_i'(0)$ and $J_i'(1)$, it follows that

- For $i \in O \cup U$, $J_i'(0) = J_i'(1)$. Therefore $0 \leq \sqrt{\frac{-2J_i'(0)}{\eta}} - \sqrt{\frac{-2J_i'(1)}{\eta}} \leq 2\lambda$.

- For $i \in V$, by the definition of $V$ and Lemma 2, it follows that $J_i'(0) = \chi_i(0)$ and $J_i'(1) = -\frac{x_i^2}{2\eta}$.

  - If $-\lambda \leq \nabla_i f(x) \leq \lambda$, then

$$\sqrt{\frac{-2J_i'(0)}{\eta}} - \sqrt{\frac{-2J_i'(1)}{\eta}} \leq \sqrt{\frac{-2J_i'(0)}{\eta}} = |\text{sign}(x_i)\lambda + \nabla_i f(x)| \leq \lambda + |\nabla_i f(x)| \leq 2\lambda.$$

  - If $\nabla_i f(x) > \lambda$, then analyzing the expressions of $J_i(\theta_i)$ in this case, there exists $\hat{\theta}_i > 1$ such that $J_i'(\hat{\theta}_i) = \chi_i(\hat{\theta}_i)$. By the non-decreasing property of $J_i'(\theta_i)$ in Corollary 1 (which can be extended to $\theta_i > 1$ trivially), $J_i'(1) \leq J_i'(\hat{\theta}_i)$. Then

$$\sqrt{\frac{-2J_i'(0)}{\eta}} - \sqrt{\frac{-2J_i'(1)}{\eta}} \le \sqrt{\frac{-2J_i'(0)}{\eta}} - \sqrt{\frac{-2J_i'(\hat\theta_i)}{\eta}}$$

$$= \quad |\text{sign}(x_i)\lambda + \nabla_i f(x)| - |\text{sign}(x_i - \hat\theta_i \eta \nabla_i f(x))\lambda + \nabla_i f(x)|$$

$$\overset{\textcircled{1}}{=} \quad (\text{sign}(x_i)\lambda + \nabla_i f(x)) - (\text{sign}(x_i - \hat\theta_i \eta \nabla_i f(x))\lambda + \nabla_i f(x))$$

$$= \quad (\text{sign}(x_i) - \text{sign}(x_i - \hat\theta_i \eta \nabla_i f(x)))\lambda$$

$$\le \quad 2\lambda,$$

where $\textcircled{1}$ is by the fact that $\nabla_i f(x) \ge \lambda$.

- If $\nabla_i f(x) < -\lambda$, we can give a similar analysis as the case $\nabla_i f(x) > \lambda$.

- If $i \in W$, by the definition of $W$ and Lemma 2, it follows that $J_i'(0) = \chi_i(0)$ and $J_i'(1) = \chi_i(1)$. Because if $i \in W$, then $|\nabla_i f(x)| \ge \lambda$, we can give a similar analysis as the case in $i \in V$.

By the above analysis, it follows that

$$\max_{i \in [d]} \left\{ \sqrt{\frac{-2J_i'(0)}{\eta}} \right\} - \max_{j \in [d]} \left\{ \sqrt{\frac{-2J_j'(1)}{\eta}} \right\} \le \max_{i \in [d]} \left\{ \sqrt{\frac{-2J_i'(0)}{\eta}} - \sqrt{\frac{-2J_i'(1)}{\eta}} \right\} \le 2\lambda.$$

In addition, by Corollary 1,

$$0 \le \max_{i \in [d]} \left\{ \sqrt{\frac{-2J_i'(0)}{\eta}} \right\} - \max_{j \in [d]} \left\{ \sqrt{\frac{-2J_j'(1)}{\eta}} \right\}.$$

Proposition 1 is proved. $\qquad\qquad\qquad\qquad\qquad\qquad\qquad\qquad\qquad\qquad\qquad\qquad\qquad\square$

## C  Proof of Lemma 3

The Lagrangian of the problem (7) is (10). By the property of KKT condition, if $(\tilde\gamma, \tilde\theta, \tilde\zeta)$ is a stationary point of the problem (10), then $\tilde\theta$ is an optimal solution of (7). Based on the value of $\tilde\theta$, one can divide $[d]$ into two disjoint parts $S$ and $T$, where

$$S = \{i : \tilde\theta_i > 0\} \text{ and } T = \{j : \tilde\theta_j = 0\}.$$

Then $\forall i \in S$, by the complementary slackness $\tilde\zeta_i \tilde\theta_i = 0$, one has $\tilde\zeta_i = 0$ and $\tilde\gamma = -J_i'(\tilde\theta_i) \ge 0$; $\forall j \in T$, similarly, one has $\tilde\zeta_j \ge 0$ and $\tilde\gamma \ge -J_j'(\tilde\theta_j) \ge 0$. Thus the KKT condition can be equivalently written as (12).

## D  Proof of Lemma 4

*Proof of Lemma 4.* By Lemma 2, it follows that $J_t'(0) \le 0$. Combing with the condition $J_s'(0) < J_t'(0)$, we have $J_s'(0) < 0$ and thus $J_s'(\theta_s)$ belongs to (case b), (case c) or (case d). Denote

$$r_s = \begin{cases} 1, & J_s'(\theta_s) \text{ belongs to } \text{(case b)}; \\ r_{s1}, & J_s'(\theta_s) \text{ belongs to } \text{(case c) or (case d)}, \end{cases} \tag{D.1}$$

where by the definition of $r_{i1}$ in Lemma 2, $r_{s1} = \frac{|x_s|}{\sqrt{-2\eta J_s'(0)}}$.

Assume by contradiction that $\tilde\theta_s = 0$ yet $\tilde\theta_t > 0$. Let $\hat\theta$ be a vector of which the elements are equal to the elements of $\tilde\theta$ except that

$$\hat\theta_s = \min\{\tilde\theta_t, r_s\}; \tag{D.2}$$

$$\hat\theta_t = \max\{0, \tilde\theta_s - r_s\}. \tag{D.3}$$

By the definition of $\hat{\theta}_s, \hat{\theta}_t$ in (D.2) and (D.3), it follows that

$$\forall \theta_s \in [\tilde{\theta}_s, \hat{\theta}_s], \qquad J_s'(\theta_s) = J_s'(0) \tag{D.4}$$

$$\forall \theta_t \in [\hat{\theta}_t, \tilde{\theta}_t], \qquad J_t'(\theta_t) \geq J_t'(0). \tag{D.5}$$

Then

$$
\begin{aligned}
J(\tilde{\theta}) - J(\hat{\theta}) &= J_s(0) + \int_0^{\tilde{\theta}_s} J_s'(\theta_s)d\theta_s + J_t(0) + \int_0^{\tilde{\theta}_t} J_t'(\theta_t)d\theta_t \\
&\quad - J_s(0) - \int_0^{\hat{\theta}_s} J_s'(\theta_s)d\theta_s - J_t(0) - \int_0^{\hat{\theta}_t} J_t'(\theta_t)d\theta_t \\
&= \int_{\hat{\theta}_s}^{\tilde{\theta}_s} J_s'(\theta_s)d\theta_s + \int_{\hat{\theta}_t}^{\tilde{\theta}_t} J_t'(\theta_t)d\theta_t \\
&\geq \int_{\hat{\theta}_s}^{\tilde{\theta}_s} J_s'(0)d\theta_s + \int_{\hat{\theta}_t}^{\tilde{\theta}_t} J_t'(0)d\theta_t \\
&\geq J_s'(0)(\tilde{\theta}_s - \hat{\theta}_s) + J_t'(0)(\tilde{\theta}_t - \hat{\theta}_t)
\end{aligned}
$$

Then by the expressions of $\hat{\theta}_s, \hat{\theta}_t$ in (D.2) and (D.3),

$$
J(\tilde{\theta}) - J(\hat{\theta}) = \begin{cases} (J_t'(0) - J_s'(0)) \cdot \hat{\theta}_t, & \hat{\theta}_t < r_s \\ (J_t'(0) - J_s'(0)) \cdot r_s, & \hat{\theta}_t \geq r_s \end{cases}. \tag{D.6}
$$

By the assumption $J_s'(0) < J_t'(0)$, $J(\tilde{\theta}) - J(\hat{\theta}) > 0$, which contradicts the fact that $\tilde{\theta}$ is the optimal solution. $\qquad\square$

# E  Proof of Lemma 5

*Proof of Lemma 5.* By the KKT condition (12) in Lemma 3, it follows that for all $i \in S$, $-J_i'(\tilde{\theta}_i) \geq \max_{j \in T} -J_j'(0)$; meanwhile by Corollary 1, for all $i \in [d]$, $-J_i'(\tilde{\theta}_i)$ is a non-increasing function. Therefore combing the KKT condition (12), we have

$$\forall i \in S, \qquad -J_i'(0) \geq -J_i'(\tilde{\theta}_i) \geq \max_{j \in T}\{-J_j'(0)\} \geq \max_{j \in T}\{-J_j'(1)\}. \tag{E.1}$$

In addition, by the KKT condition (12), for all $i_1 \in S, i_2 \in S$, $-J_{i_1}'(\tilde{\theta}_{i_1}) = -J_{i_2}'(\tilde{\theta}_{i_2})$. Because by Corollary 1, for all $i \in [d]$, $-J_i'(\tilde{\theta}_i)$ is a non-increasing function, therefore

$$\forall i_1 \in S, i_2 \in S, \qquad -J_{i_1}'(0) \geq -J_{i_1}'(\tilde{\theta}_{i_1}) = -J_{i_2}'(\tilde{\theta}_{i_2}) \geq -J_{i_2}'(1).$$

Therefore it follows that

$$\forall i \in S, \qquad -J_i'(0) \geq \max_{j \in S} -J_j'(1). \tag{E.2}$$

By combing (E.1) and (E.2), we get

$$-J_i'(0) \geq \max_{j \in [d]} -J_j'(1). \tag{E.3}$$

$\qquad\square$

# F  Proof of Theorem 1

*Proof.* To prove Theorem 1, by Lemma 1, we only need to show $\tilde{\theta}$ in Alg. 1 is the optimal solution of the problem (7). By Lemma 3, to prove the optimality of $\tilde{\theta}$ in Alg. 1, we only need to show the $\tilde{\gamma}, \tilde{\theta}$ in Alg. 1 satisfy the KKT condition in Lemma 3. Equivalently, we rewrite the KKT condition as follows,

$$
\begin{cases}
\sum_{i \in [d]} \tilde{\theta}_i = 1, & \text{(F.1a)} \\[2mm]
\text{for all } i \in [d], \quad \tilde{\theta}_i \geq 0, & \text{(F.1b)} \\[2mm]
\text{for all } i \in S, \quad \tilde{\gamma} = -J_i'(\tilde{\theta}_i) \geq \max_{j \in T} -J_j'(0), & \text{(F.1c)}
\end{cases}
$$

where as in Lemma 3, $S = \{i \in [d] | \tilde{\theta}_i > 0\}, T = \{i \in [d] | \tilde{\theta}_i = 0\}$.

By checking the step 5 in Alg. 1, it is found that (F.1a) is already satisfied. Meanwhile, for $i \in R_\rho \backslash \{i_\rho\}$ and $i \in [d] \backslash R_\rho$, $\tilde{\theta}_i \geq 0$. So the remaining work is to show that the two conditions $\tilde{\theta}_{i_\rho} \geq 0$ and (F.1c) can be satisfied, which is given in Lemmas 6, 7 and 8.

**Lemma 6.** *Assume $\tilde{\gamma}$ is defined in the step 4 of Alg. 1, then $\tilde{\gamma} \geq \max_{j \in [d]} -J_j'(0)$.*

**Lemma 7.** *As in the step 5 of Alg. 1, for all $l \in R_\rho \backslash \{i_\rho\}$, by setting $\tilde{\theta}_l = \frac{|x_l|}{\sqrt{2\eta\tilde{\gamma}}}$, it follows that $\tilde{\gamma} = -J_l'(\tilde{\theta}_l)$.*

**Lemma 8.** *As in the step 5 of Alg. 1, by setting $\tilde{\theta}_{i_\rho} = 1 - \sum_{k \in R_\rho \backslash \{i_\rho\}} \tilde{\theta}_k$, then it follows that $\tilde{\theta}_{i_\rho} \geq 0$ always hold. Meanwhile, if $\tilde{\theta}_{i_\rho} > 0$, then $\tilde{\gamma} = -J_{i_\rho}'(\tilde{\theta}_{i_\rho})$.*

By Lemma 8, $\tilde{\theta}_{i_\rho} \geq 0$. To show (F.1c) can be satisfied, we give the following discussion.

- If $\tilde{\theta}_{i_\rho} = 0$, let $S = R_\rho \backslash \{i_\rho\}$ and $T = [d] \backslash (R_\rho \backslash \{i_\rho\})$, then by Lemmas 6, 7 and 8, (F.1c) is satisfied.

- If $\tilde{\theta}_{i_\rho} > 0$, let $S = R_\rho$ and $T = [d] \backslash R_\rho$, then by Lemmas 6, 7 and 8, (F.1c) is satisfied.

Therefore Theorem 1 is proved. $\qquad\square$

## G  Technical Lemmas and Proofs of Lemmas 6, 7 and 8

The main difficulty in the proof of Lemmas 7 and 8 comes from the fact that by Lemma 2, for all $i \in [d]$ and $0 \leq \theta_i \leq 1$, the expression of $J_i'(\theta_i)$ has 4 different cases. Here we give Lemma 9 to show an equivalence relation between the expression of $J_i'(\theta_i)$ and the relation of $J_i'(0)$ and $J_i'(1)$.

**Lemma 9.** *For all $i \in [d]$ and $0 \leq \theta_i \leq 1$, $J_i'(\theta_i)$ belongs to the (case a) or (case b) in Lemma 2 if and only if $J_i'(0) = J_i'(1)$; $J_i'(\theta_i)$ belongs to the (case c) or (case d) in Lemma 2 if and only if $J_i'(0) \neq J_i'(1)$.*

*Proof of Lemma 9.* For all $i \in [d]$ and $0 \leq \theta_i \leq 1$, by observing the (case a), (case b), (case c) and (case d) of $J_i'(\theta_i)$ in Lemma 2, it follows that $J_i'(\theta_i)$ belongs to the (case a) or (case b) if and only if it is a constant function, which implies $J_i'(0) = J_i'(1)$. $J_i'(\theta_i)$ belongs to the (case c) or (case d) if and only if it is a piecewise function, which implies $J_i'(0) \neq J_i'(1)$. $\qquad\square$

By Lemma 9, the condition $J_{i_j}'(0) = J_{i_j}'(1)$ in the step 3 of Alg. 1 is used to identify which case $J_{i_j}'(\theta_{i_j})$ belongs to. Lemma 10 introduces an implied result of the conditions in the step 3 of Alg. 1. (In the following lemmas, we assume that $r_{i1}, r_{i2}$ $(i \in [d])$ have been defined in Lemma 2 and $i_m, v_m, Q, v, \rho, \tilde{\gamma}, i_j, R_j (j \in [|Q| + 1])$ have been defined in Alg. 1.)

**Lemma 10.** *For all $j \in [\rho - 1]$, it follows that all the following conditions*

$$
\begin{cases}
J_{i_j}'(0) \neq J_{i_j}'(1), & \text{(G.1a)} \\[2mm]
\displaystyle\sum_{l \in R_j} |x_l| < \sqrt{2\eta v_j}, & \text{(G.1b)} \\[2mm]
j < |Q| + 1, & \text{(G.1c)}
\end{cases}
$$

*must be satisfied.*

*Proof of Lemma 10.* By the step 3 in Alg. 1, $\rho$ is the minimal index that can satisfy one of the 3 following conditions

$$
\begin{cases}
J_{i_j}'(0) = J_{i_j}'(1), \\[2mm]
\displaystyle\sum_{l \in R_\rho} |x_l| \geq \sqrt{2\eta v_\rho}, \\[2mm]
\rho = |Q| + 1,
\end{cases}
$$

which implies that for all $j \in [\rho - 1]$, $j$ satisfies all the 3 conditions in Lemma 10. $\qquad \square$

By Lemma 10, $j \in [\rho - 1]$ shares the 3 common properties in (G.1a)-(G.1c), which is important for the proof of the subsequent lemmas about $j \in [\rho - 1]$. In Lemma 11, we can find useful inequalities.

**Lemma 11.** *For all $j \in [\rho - 1]$, $v_j \geq \tilde{\gamma} \geq v_\rho \geq v_m \geq -J'_{i_j}(1)$.*

*Proof of Lemma 11.* By the step 4 in Alg. 1, $\tilde{\gamma}$ has two possible values.

If $\tilde{\gamma} = (\sum_{k \in R_{\rho-1}} |x_k|)^2/(2\eta)$, it follows that

- In Lemma 10, let $j = \rho - 1$, we have $\sum_{k \in R_{\rho-1}} |x_k| < \sqrt{2\eta v_{\rho-1}}$. Then $v_{\rho-1} \geq \tilde{\gamma}$. Then by the definition of $v$, $v_1 \geq v_2 \geq \cdots \geq v_{\rho-1}$. Thus for all $j \in [\rho - 1]$, we have $v_j \geq \tilde{\gamma}$.

- In the step 4, when $\tilde{\gamma} = (\sum_{k \in R_{\rho-1}} |x_k|)^2/(2\eta)$, by the condition $\sum_{k \in R_{\rho-1}} |x_k| \geq \sqrt{2\eta v_\rho}$, we have $\tilde{\gamma} \geq v_\rho$.

If $\tilde{\gamma} = v_\rho$, by the definition of $v$, $v_1 \geq v_2 \geq \cdots \geq v_\rho$. Then it follows that for all $j \in [\rho - 1]$, $v_j \geq \tilde{\gamma} \geq v_\rho$.

By the definition of $v$, $v_1 \geq v_2 \geq \cdots \geq v_\rho \geq \cdots \geq v_m$. By the definition of $v_m$, for all $j \in [\rho - 1]$, $v_m = \max_{i \in [d]} -J'_i(1) \geq -J'_{i_j}(1)$. Therefore $v_\rho \geq v_m \geq -J'_{i_j}(1)$.

Combining the above analysis, Lemma 11 is proved.

$\qquad \square$

Lemma 11 gives $\tilde{\gamma}$ both lower and upper bounds, which then further bounds the range of $\tilde{\theta}_l = \frac{|x_l|}{\sqrt{2\eta\tilde{\gamma}}}$.

Before continue, we show the relation between $R_{\rho-1}$ and $R_\rho \backslash \{i_\rho\}$.

**Lemma 12.** *If $\rho < |Q| + 1$, then $R_{\rho-1} = R_\rho \backslash \{i_\rho\}$; if $v_{|Q|+1} = -J'_{i_{|Q|+1}}(1)$, then $R_{|Q|} = R_{|Q|+1} \backslash \{i_{|Q|+1}\}$; if $v_{|Q|+1} \neq -J'_{i_{|Q|+1}}(1)$, then $R_{|Q|} = R_{|Q|+1}$.*

*Proof of Lemma 12.*

- If $\rho < |Q| + 1$, then by the step 2 in Alg. 1, $i_1, i_2, \ldots, i_{|Q|}$ are different coordinates. Thus $R_{\rho-1} = R_\rho \backslash \{i_\rho\}$.

- If $v_{|Q|+1} = -J'_{i_{|Q|+1}}(1)$, by the definition of $Q$ in the step 1 of Alg. 1, $i_{|Q|+1} \notin Q$. Therefore $R_{|Q|} = R_{|Q|+1} \backslash \{i_{|Q|+1}\}$.

- If $v_{|Q|+1} \neq -J'_{i_{|Q|+1}}(1)$, then by Lemma 9, $J'_{i_{|Q|+1}}(\theta_{i_{|Q|+1}})$ belongs to (case c) or (case d). It follows that $-J'_{i_{|Q|+1}}(0) > -J'_{i_{|Q|+1}}(1) = v_m$. Then by the definition of $|Q|$ in the step 1 of Alg. 1, $i_{|Q|+1} \in Q$. Therefore $R_{|Q|} = R_{|Q|+1}$.

$\qquad \square$

Based on the above technical lemmas, we can prove Lemmas 6, 7 and 8.

### G.1 Proofs of Lemma 6, 7 and 8

*Proof of Lemma 6.* By the step 4 of Alg. 1, if $\sum_{l \in R_{\rho-1}} |x_l| \geq \sqrt{2\eta v_\rho}$, then $\tilde{\gamma} = (\sum_{k \in R_{\rho-1}} |x_k|)^2/(2\eta)$ which means $\tilde{\gamma} \geq v_\rho$; otherwise, $\tilde{\gamma} = v_\rho$.

In addition by the definition of $v$, it follows that $v_1 \geq v_2 \geq \cdots \geq v_\rho \geq \cdots \geq v_m = \max_{j \in [d]} -J'_j(0)$.

Therefore $\tilde{\gamma} \geq v_\rho \geq \max_{j \in [d]} -J'_j(0)$. Lemma 6 is proved.

$\qquad \square$

*Proof of Lemma 7.* By Lemma 12, $R_\rho \backslash \{i_\rho\} \subset R_{\rho-1}$. Then by Lemma 10, for all $l \in R_\rho \backslash \{i_\rho\}$, $J_l'(0) \neq J_l'(1)$. Then by Lemma 9, for all $l \in R_\rho \backslash \{i_\rho\}$ and $0 \leq \theta_l \leq 1$, $J_l'(\theta_l)$ belongs to (case c) or (case d) in Lemma 2.

Assume that for $i \in [d]$, $r_{i1} = \frac{|x_i|}{\sqrt{-2\eta J_i'(0)}}$ and $r_{i2} = \frac{|x_i|}{\sqrt{-2\eta J_i'(1)}}$ have been defined in Lemma 2. By Lemma 11, we have $-J_{i_j}'(0) = v_j \geq \tilde{\gamma} \geq -J_{i_j}'(1)$. Then $r_{i1} \leq \tilde{\theta}_l = \frac{|x_i|}{\sqrt{2\eta\tilde{\gamma}}} \leq r_{i2}$.

In addition, if $\tilde{\gamma} = (\sum_{k \in R_{\rho-1}} |x_k|)^2/(2\eta)$, when $\tilde{\theta}_l = \frac{|x_l|}{\sum_{k \in R_{\rho-1}} |x_k|} \leq 1$; if $\tilde{\gamma} = v_\rho$, by the step 4 in Alg. 1, the condition $\sum_{k \in R_{\rho-1}} |x_k| \leq \sqrt{2\eta v_\rho}$ holds. Then $\tilde{\theta}_l = \frac{|x_k|}{\sqrt{2\eta\tilde{\gamma}}} = \frac{|x_k|}{\sqrt{2\eta v_\rho}} \leq 1$.

Therefore $r_{i1} \leq \tilde{\theta}_l \leq \min\{r_{i2}, 1\}$. By the form of (case c) and (case d) in Lemma 2, we can find that for all $l \in R_\rho \backslash \{i_\rho\}$, $J_l'(\tilde{\theta}_l) = -\frac{|x_l|^2}{2\eta\theta_l^2} = -\tilde{\gamma}$.

$\square$

To prove the KKT condition (F.1c), besides Lemma 7 for the case $j \in [\rho - 1]$, we also need Lemma 8 for the case $j = \rho$.

*Proof of Lemma 8.* By the step 4 in Alg. 1, $\tilde{\gamma}$ has two possible values.

If $\tilde{\gamma} = (\sum_{k \in R_{\rho-1}} |x_k|)^2/(2\eta)$, by analyzing the following 3 cases, we can show that $\tilde{\theta}_{i_\rho} \geq 0$ always holds and if $\tilde{\theta}_{i_\rho} > 0$, then $J_{i_\rho}'(\tilde{\theta}_{i_\rho}) = -\tilde{\gamma}$.

- If $\rho < |Q| + 1$, then by Lemma 10, $R_\rho \backslash \{i_\rho\} = R_{\rho-1}$. By the step 5 in Alg. 1 and $\tilde{\gamma} = (\sum_{k \in R_{\rho-1}} |x_k|)^2/(2\eta)$, for all $l \in R_\rho \backslash \{i_\rho\}$, $\tilde{\theta}_l = \frac{|x_l|}{\sqrt{2\eta\tilde{\gamma}}} = \frac{|x_l|}{\sum_{k \in R_{\rho-1}} |x_k|}$. Then $\sum_{l \in R_\rho \backslash \{i_\rho\}} \tilde{\theta}_l = \sum_{l \in R_{\rho-1}} \tilde{\theta}_l = 1$. Therefore $\tilde{\theta}_{i_\rho} = 1 - \sum_{k \in R_\rho \backslash \{i_\rho\}} \tilde{\theta}_k = 0$.

- If $\rho = |Q| + 1$ and $J_{i_\rho}'(0) = J_{i_\rho}'(1)$, then by Lemma 10, it follows that $R_\rho \backslash \{i_\rho\} = R_{\rho-1}$. Therefore, by the same analysis in the case $\rho < |Q| + 1$, $\tilde{\theta}_{i_\rho} = 1 - \sum_{k \in R_\rho \backslash \{i_\rho\}} \tilde{\theta}_k = 0$.

- If $\rho = |Q| + 1$ and $J_{i_\rho}'(0) \neq J_{i_\rho}'(1)$, then by Lemma 10, $R_{\rho-1} = R_\rho = (R_\rho \backslash \{i_\rho\}) \cup \{i_\rho\}$. Therefore $0 \leq \tilde{\theta}_{i_\rho} = 1 - \sum_{k \in R_\rho \backslash \{i_\rho\}} \tilde{\theta}_k = \frac{|x_{i_\rho}|}{\sum_{k \in R_{\rho-1}} |x_k|} \leq 1$. Meanwhile, by Lemma 9, $J_{i_\rho}'(\tilde{\theta}_{i_\rho})$ belongs to (case c) or (case d) in Lemma 2. Due to $\tilde{\gamma} = (\sum_{k \in R_{\rho-1}} |x_k|)^2/(2\eta)$, then the condition $\sum_{k \in R_{\rho-1}} |x_k| \geq \sqrt{2\eta v_\rho}$ holds. Thus

$$\tilde{\theta}_{i_\rho} \leq \frac{|x_{i_\rho}|}{\sqrt{2\eta v_\rho}} = \frac{|x_{i_\rho}|}{\sqrt{2\eta v_{|Q|+1}}} = \frac{|x_{i_\rho}|}{\sqrt{-2\eta J_{i_\rho}'(1)}} = r_{i_\rho 2},$$

where $r_{i_\rho 2}$ is defined in Lemma 2. Meanwhile, in Lemma 10, let $j = \rho - 1$, then $\sum_{k \in R_{\rho-1}} |x_k| < \sqrt{2\eta v_{\rho-1}}$. Thus $\tilde{\theta}_{i_\rho} > \frac{|x_{i_\rho}|}{\sqrt{2\eta v_{\rho-1}}} = \frac{|x_{i_\rho}|}{\sqrt{2\eta v_{|Q|}}}$. In addition, due to $i_\rho \in Q$, we have $v_{i_\rho} \geq v_{|Q|}$. Thus

$$\tilde{\theta}_{i_\rho} > \frac{|x_{i_\rho}|}{\sqrt{2\eta v_{i_\rho}}} = \frac{|x_{i_\rho}|}{\sqrt{-2\eta J_{i_\rho}'(0)}} = r_{i_\rho 1},$$

where $r_{i_\rho 1}$ is defined in Lemma 2. Combing the above analyses, we have

$$0 < r_{i_\rho 1} < \tilde{\theta}_{i_\rho} \leq \min\{1, r_{i_\rho 2}\}.$$

Therefore by the form of (case c) and (case d) in Lemma 2, $J_{i_\rho}'(\tilde{\theta}_{i_\rho}) = -\frac{x_{i_\rho}^2}{2\eta\tilde{\theta}_{i_\rho}} = -\tilde{\gamma}$.

If $\tilde{\gamma} = v_\rho$, according to the condition in the step 4 of Alg. 1, $\sum_{l \in R_{\rho-1}} |x_l| < \sqrt{2\eta v_\rho}$. Meanwhile by Lemma 12, we have $R_{i_\rho} \backslash \{i_\rho\} \subset R_{i_{\rho-1}}$. Then $\sum_{l \in R_{i_\rho} \backslash \{i_\rho\}} \tilde{\theta}_l = \sum_{l \in R_{i_\rho} \backslash \{i_\rho\}} \frac{|x_l|}{\sqrt{2\eta\tilde{\gamma}}} \leq \sum_{l \in R_{i_{\rho-1}}} \frac{|x_l|}{\sqrt{2\eta\tilde{\gamma}}} < 1$. Therefore $\tilde{\theta}_{i_\rho} = 1 - \sum_{l \in R_{i_\rho} \backslash \{i_\rho\}} \tilde{\theta}_l > 0$. We can give the analyses by discussing the following 3 cases.

- If $v_\rho = -J'_{i_\rho}(1)$, then by Lemma 9, $J_{i_\rho}(\theta_{i_\rho})$ belongs to (case a) or (case b). Therefore $J'_{i_\rho}(\tilde{\theta}_{i_\rho}) = J'_{i_\rho}(0) = -v_\rho = -\tilde{\gamma}$.

- If $v_\rho \neq -J'_{i_\rho}(1)$ and $\sum_{l \in R_\rho} |x_l| \geq \sqrt{2\eta v_\rho}$, then by Lemma 9, $J'_{i_\rho}(\theta_{i_\rho})$ belongs to (case c) or (case d) in Lemma 2. Meanwhile for $l \in R_\rho \backslash \{i_\rho\}$, by $\tilde{\theta}_l = \frac{|x_l|}{\sqrt{2\eta\tilde{\gamma}}} = \frac{|x_l|}{\sqrt{2\eta v_\rho}}$, we have

$$\tilde{\theta}_{i_\rho} = 1 - \sum_{l \in R_\rho \backslash \{i_\rho\}} \tilde{\theta}_l = 1 - \sum_{l \in R_\rho \backslash \{i_\rho\}} \frac{|x_l|}{\sqrt{2\eta v_\rho}} \leq \frac{|x_{i_\rho}|}{\sqrt{2\eta v_\rho}} = r_{i_\rho 1},$$

where $r_{i_\rho 1}$ is defined in Lemma 2 and the last inequality is due to the condition $\sum_{l \in R_\rho} |x_l| \geq \sqrt{2\eta v_\rho}$. Therefore by the form of (case c) and (case d) in Lemma 2, it follows that $J'_{i_\rho}(\tilde{\theta}_{i_\rho}) = J'_{i_\rho}(0) = -v_\rho = -\tilde{\gamma}$.

- If $v_\rho \neq -J'_{i_\rho}(1)$, $\sum_{l \in R_\rho} |x_l| < \sqrt{2\eta v_\rho}$ and $\rho = |Q| + 1$, then by $v_\rho \neq -J'_{i_\rho}(1)$ and Lemma 9, $J'_{i_\rho}(\theta_{i_\rho})$ belongs to (case c) or (case d) in Lemma 2. By $\sum_{l \in R_\rho} |x_l| < \sqrt{2\eta v_\rho}$, we have that the $r_{i_\rho 2}$ in Lemma 2 satisfies

$$r_{i_\rho 2} = \frac{|x_{i_\rho}|}{\sqrt{-2\eta J'_{i_\rho}(1)}} = \frac{|x_{i_\rho}|}{\sqrt{2\eta v_{|Q|+1}}} = \frac{|x_{i_\rho}|}{\sqrt{2\eta v_\rho}} < \frac{\sum_{l \in R_\rho} |x_l|}{\sqrt{2\eta v_\rho}} < 1,$$

therefore $J'_{i_\rho}(\theta_{i_\rho})$ belongs to the (case d) in Lemma 2. For all $l \in R_\rho \backslash \{i_\rho\}$, by $\tilde{\theta}_l = \frac{|x_l|}{\sqrt{2\eta\tilde{\gamma}}} = \frac{|x_l|}{\sqrt{2\eta v_\rho}}$ and $\sum_{l \in R_\rho} |x_l| < \sqrt{2\eta v_\rho}$, we have

$$\tilde{\theta}_{i_\rho} = 1 - \sum_{k \in R_\rho \backslash \{i_\rho\}} \tilde{\theta}_k \geq \frac{|x_{i_\rho}|}{\sqrt{2\eta v_\rho}} = r_{i_\rho 2}. \tag{G.3}$$

Therefore, by the form of the (case d) in Lemma 2, $J'_{i_\rho}(\tilde{\theta}_{i_\rho}) = J'_{i_\rho}(1) = -v_\rho = -\tilde{\gamma}$.

Summarizing the above analyses, we prove Lemma 8.

$\square$

## H  Proof of Theorem 2

### H.1  Some necessary Lemmas and Definitions

For $1 < p < \infty$ and the $\ell_p$-norm $\| \cdot \|_p$, we denote its dual norm as $\|x\|_q = \max_{\|y\|_p \leq 1} x^T y = (\sum_{i=1}^d |x_i|^q)^{\frac{1}{q}}$, where $\frac{1}{p} + \frac{1}{q} = 1$. For $p = 1$, by the definition of dual norm, the dual norm of $\ell_1$-norm is $\ell_\infty$-norm. In Lemma 13 and 14, some classical results are described.

**Lemma 13.** *([8, §3.1.9]) If $\forall x, y \in \mathbb{R}^d, 1 \leq p \leq \infty, \frac{1}{p} + \frac{1}{q} = 1$, and $\eta > 0$, then $|\langle x, y \rangle| \leq \|x\|_q \|y\|_p \leq \frac{1}{2\eta} \|x\|_q^2 + \frac{\eta}{2} \|y\|_p^2$.*

**Lemma 14.** *If $\forall x \in \mathbb{R}^d, 1 \leq p \leq \infty$, then $\|x\|_p \leq \|x\|_1 \leq n^{1-\frac{1}{p}} \|x\|_p$.*

For a continuous differentiable function $f(x)$, we give the following definitions.

**Definition 1.** *$f(x)$ is $L_p$-smooth ($1 \leq p \leq \infty$) w.r.t. $\| \cdot \|_p$ if $\forall x, y \in \mathbb{R}^d$ and $\frac{1}{p} + \frac{1}{q} = 1$, $\|\nabla f(x) - \nabla f(y)\|_q \leq L_p \|x - y\|_p$.*

By Definition 1, we have Lemma 15.

**Lemma 15.** *If $f(x)$ is $L_p$-smooth $(1 \leq p \leq \infty)$ w.r.t $\|\cdot\|_p$ and $\frac{1}{p} + \frac{1}{q} = 1$, then*

$$\frac{1}{2L_p}\|\nabla f(x) - \nabla f(y)\|_q^2 \leq f(y) - f(x) - \langle \nabla f(x), y - x \rangle \leq \frac{L_p}{2}\|x - y\|_p^2. \tag{H.1}$$

*Proof.* Firstly it is showed that $f(x)$ being $L_p$-smooth w.r.t. $\|\cdot\|_p$ implies that $\forall x, y \in \mathbb{R}^d$

$$f(y) \leq f(x) + \langle \nabla f(x), y - x \rangle + \frac{L_p}{2}\|y - x\|_p^2.$$

Consider the function $g(\tau) = f(x + \tau(y - x))$ with $\tau \in \mathbb{R}$. Then

$$\begin{aligned}
f(y) - f(x) - \langle \nabla f(x), y - x \rangle &= g(1) - g(0) - \langle \nabla f(x), y - x \rangle \\
&= \int_0^1 \left( \frac{dg(\tau)}{d\tau} - \langle \nabla f(x), y - x \rangle \right) d\tau \\
&= \int_0^1 \left( \langle \nabla f(x + \tau(y - x)), y - x \rangle - \langle \nabla f(x), y - x \rangle \right) d\tau \\
&= \int_0^1 \langle \nabla f(x + \tau(y - x)) - \nabla f(x), y - x \rangle \, d\tau \\
&\leq \int_0^1 \|\nabla f(x + \tau(y - x)) - \nabla f(x)\|_q \|y - x\|_p \, d\tau \\
&\leq \int_0^1 L_p \tau \|y - x\|_p^2 \, d\tau \\
&= \frac{L_p}{2}\tau^2\|y - x\|_p^2 \Big|_0^1 \\
&= \frac{L_p}{2}\|y - x\|_p^2.
\end{aligned}$$

To subsequently show $\frac{1}{2L_p}\|\nabla f(x) - \nabla f(y)\|_q^2 \leq f(y) - f(x) - \langle \nabla f(x), y - x \rangle$, fix $x \in \mathbb{R}^d$ and consider the function

$$\phi(y) = f(y) - \langle \nabla f(x), y \rangle,$$

which is convex on $\mathbb{R}^d$ and also has an $L_p$-Lipschitz continuous gradient w.r.t. $\|\cdot\|_p$, as

$$\begin{aligned}
\|\phi'(y) - \phi'(x)\|_q &= \|(\nabla f(y) - \nabla f(x)) - (\nabla f(x) - \nabla f(x))\|_q \\
&= \|\nabla f(y) - \nabla f(x)\|_q \\
&\leq L_p \|y - x\|_p.
\end{aligned}$$

As the minimizer of $\phi$ is $x$ (i.e., $\phi'(x) = 0$), for any $y \in \mathbb{R}^d$, we have

$$\begin{aligned}
\phi(x) = \min_v \phi(v) &\leq \min_v \left\{ \phi(y) + \langle \phi'(y), v - y \rangle + \frac{L_p}{2}\|v - y\|_p^2 \right\} \\
&= \phi(y) - \sup_v \{ \langle -\phi'(y), v - y \rangle - \frac{L_p}{2}\|v - y\|_p^2 \} \\
&= \phi(y) - \frac{1}{2L_p}\|\phi'(y)\|_q^2.
\end{aligned}$$

Substituting in the definition of $\phi$, we have

$$f(x) - \langle \nabla f(x), x \rangle \leq f(y) - \langle \nabla f(x), y \rangle - \frac{1}{2L_p}\|\nabla f(y) - \nabla f(x)\|_q^2$$

$$\iff \qquad \frac{1}{2L_p}\|\nabla f(y) - \nabla f(x)\|_q^2 \leq f(y) - f(x) - \langle \nabla f(x), y - x \rangle.$$

$\square$

**Definition 2.** *$f(x)$ is $\sigma_p$-strongly convex ($1 \leq p \leq \infty$) w.r.t. $\|\cdot\|_p$ if $\forall x, y \in \mathbb{R}^d$ and $\frac{1}{p} + \frac{1}{q} = 1$, $f(y) - f(x) - \langle \nabla f(x), y - x \rangle \geq \frac{\sigma_p}{2} \|x - y\|_p^2$.*

Taking $\frac{1}{2}\|x\|_p^2$ ($1 < p \leq 2$) as an example. It is known that $\frac{1}{2}\|x\|_p^2$ is $(p-1)$-strongly convex *w.r.t.* $\|\cdot\|_p$ [6]. Based on $\frac{1}{2}\|x\|_p^2$ ($1 < p \leq 2$), one can define $p$-Bregman divergence

$$B_p(y, x) = \frac{1}{2}\|y\|_p^2 - \frac{1}{2}\|x\|_p^2 - \langle \nabla \frac{1}{2}\|x\|_p^2, y - x \rangle. \tag{H.2}$$

**Lemma 16** ([6, 2]). *For $x, y \in \mathbb{R}^d$, $1 < p \leq 2$, $B_p(y, x) = \frac{1}{2}\|y\|_p^2 - \frac{1}{2}\|x\|_p^2 - \langle \nabla \frac{1}{2}\|x\|_p^2, y - x \rangle$ satisfies the 3 properties.*

- $B_p(y, x) \geq \frac{p-1}{2}\|y - x\|_p^2$;

- $B_p(y, x) = 0$ if and only if $y = x$;

- $B_p(x, y) + B_p(y, z) = B_p(x, z) + \langle \frac{1}{2}\nabla\|z\|_p^2 - \frac{1}{2}\nabla\|y\|_p^2, x - y \rangle$.

## H.2 Proof of Theorem 2

Theorem 2 is proved by following the steps of the proof in [1]. First, in Section H.2.1, ASGCD is analyzed for the fixed $k$-th iteration. In the one-iteration analysis, $y_k$, $z_k$ and $x_{k+1}$ are assumed to be fixed and thus the selection of the mini batch $\mathcal{B}$ in the $k$-th iteration is the only source of randomness. For simplicity, let $\tilde{x} = \tilde{x}^s$, $\tau_1 = \tau_{1,s}$, $\alpha = \alpha_s$ where $s = \lfloor \frac{k}{m} \rfloor$ is the epoch corresponding to $k$. Let $\beta(b) \overset{\text{def}}{=} \frac{n-b}{b(n-1)}$ and denote $\sigma_{k+1}^2 \overset{\text{def}}{=} \|\nabla f(x_{k+1}) - \tilde{\nabla}_{k+1}\|_\infty^2$. Then $\mathbb{E}[\sigma_{k+1}^2]$ is the variance measured by $\|\cdot\|_\infty$ of the gradient estimator $\tilde{\nabla}_{k+1}$ in this iteration. Second, in Section H.2.2, Theorem 2 is proven by combing the one-iteration analysis in Section H.2.1 into the outer-iteration analysis in Section H.2.2.

There are 3 differences from the analysis in [1]. First, the analysis is used for the specific ASGCD algorithm that combines SOTOPO and pCOMID and thus the value of the parameter $\alpha_s$ is different from the setting in [1]. Second, the analysis is given under the mini batch selection setting and $\|\cdot\|_\infty$ rather than one sample selection setting and $\|\cdot\|_2$. Third, we use a different way to represent the convergence result for $\ell_1$-regularized problems (1).

### H.2.1 One-iteration analysis

**Lemma 17** (SOTOPO). *If*

$$
\begin{aligned}
y_{k+1} &= SOTOPO(\tilde{\nabla}_{k+1}, x_{k+1}, \lambda, \eta), \quad \text{and} \\
Prog(x_{k+1}) &= -\min_{y \in \mathbb{R}^d}\left\{ \frac{(1 + 2\beta(b))L_1}{2}\|y - x_{k+1}\|_1^2 + \langle \tilde{\nabla}_{k+1}, y - x_{k+1} \rangle + \lambda\|y\|_1 - \lambda\|x_{k+1}\|_1 \right\} \geq 0,
\end{aligned}
$$

*it follows that if $b < n$ (where the expectation is only over the randomness of $\tilde{\nabla}_{k+1}$), then*

$$F(x_{k+1}) - \mathbb{E}[F(y_{k+1})] \geq \mathbb{E}[Prog(x_{k+1})] - \frac{1}{4\beta(b)L_1}\mathbb{E}[\sigma_{k+1}^2]; \tag{H.3}$$

*if $b = n$ (no randomness exists), then*

$$F(x_{k+1}) - \mathbb{E}[F(y_{k+1})] \geq \mathbb{E}[Prog(x_{k+1})]. \tag{H.4}$$

*Proof.* If $b < n$, it follows that

$$
\begin{aligned}
\text{Prog}(x_{k+1}) \quad = \quad & -\min_{y \in \mathbb{R}^d} \left\{ \frac{(1 + 2\beta(b)) L_1}{2} \|y - x_{k+1}\|_1^2 + \langle \tilde{\nabla}_{k+1}, y - x_{k+1} \rangle + \lambda \|y\|_1 - \lambda \|x_{k+1}\|_1 \right\} \\
\overset{\text{①}}{=} \quad & -\left( \frac{(1 + 2\beta(b)) L_1}{2} \|y_{k+1} - x_{k+1}\|_1^2 + \langle \tilde{\nabla}_{k+1}, y_{k+1} - x_{k+1} \rangle + \lambda \|y_{k+1}\|_1 - \lambda \|x_{k+1}\|_1 \right) \\
= \quad & -\left( \frac{L_1}{2} \|y_{k+1} - x_{k+1}\|_1^2 + \langle \nabla f(x_{k+1}), y_{k+1} - x_{k+1} \rangle + \lambda \|y_{k+1}\|_1 - \lambda \|x_{k+1}\|_1 \right) \\
& + \left( \langle \nabla f(x_{k+1}) - \tilde{\nabla}_{k+1}, y_{k+1} - x_{k+1} \rangle - \beta(b) L_1 \|y_{k+1} - x_{k+1}\|_1^2 \right) \\
\overset{\text{②}}{\leq} \quad & -(f(y_{k+1}) - f(x_{k+1}) + \lambda \|y_{k+1}\|_1 - \lambda \|x_{k+1}\|_1) + \frac{1}{4\beta(b) L_1} \|\nabla f(x_{k+1}) - \tilde{\nabla}_{k+1}\|_\infty^2,
\end{aligned}
$$

where ① is by Theorem 1, ② is by the smoothness assumption (2), Lemma 13 and 15. Taking expectation on both sides, (H.3) is obtained.

If $b = n$, then $\beta(b) = 0$. By using a similar analysis as the case $b < n$, (H.4) is obtained. $\qquad\square$

**Lemma 18.** *(variance upper bound). If $b < n$, then*

$$
\mathbb{E}[\|\tilde{\nabla}_{k+1} - \nabla f(x_{k+1})\|_\infty^2] \leq 2\beta(b) L_1 (f(\tilde{x}) - f(x_{k+1}) - \langle \nabla f(x_{k+1}), \tilde{x} - x_{k+1} \rangle). \qquad \text{(H.5)}
$$

*Proof.* Before the proof, it should be noted that the variance upper bound measured by $\|\cdot\|_2$ of mini-batch selection has been proved in [27]. The variance in our case is measured by $\|\cdot\|_\infty$. Because some properties of $\|\cdot\|_2$ such as $\mathbb{E}[\|x - \mathbb{E}[x]\|_2^2] = \mathbb{E}[\|x\|_2^2] - \|\mathbb{E}[x]\|_2^2$ and $\|\sum_i x_i\|_2^2 = \sum_{i,j} x_i^T x_j$ can't be generalized to $\|\cdot\|_\infty$ directly, the proof is slightly different from the proof in [27].

Let $\phi_j = (\nabla f_j(x_{k+1}) - \nabla f_j(\tilde{x})) - (\nabla f(x_{k+1}) - \nabla f(\tilde{x}))$ and $\phi_j^i = (\nabla_i f_j(x_{k+1}) - \nabla_i f_j(\tilde{x})) - (\nabla_i f(x_{k+1}) - \nabla_i f(\tilde{x}))$. Denote $i_{\max} = \arg\max_i |\nabla_i f(x_{k+1}) - \tilde{\nabla}_{k+1,i}|$. It follows that

$$
\begin{aligned}
\mathbb{E}\left[ \left\| \frac{1}{b} \sum_{j \in \mathcal{B}} \phi_j^{i_{\max}} \right\|_\infty^2 \right] \quad = \quad & \frac{1}{b^2} \mathbb{E}\left[ \sum_{j_1, j_2 \in \mathcal{B}} \phi_{j_1}^{i_{\max}} \phi_{j_2}^{i_{\max}} \right] \\
= \quad & \frac{1}{b^2} \mathbb{E}\left[ \sum_{j_1 \neq j_2 \in \mathcal{B}} \phi_{j_1}^{i_{\max}} \phi_{j_2}^{i_{\max}} \right] + \frac{1}{b} \mathbb{E}\left[ (\phi_j^{i_{\max}})^2 \right] \\
= \quad & \frac{b-1}{bn(n-1)} \sum_{j_1 \neq j_2 \in [n]} \phi_{j_1}^{i_{\max}} \phi_{j_2}^{i_{\max}} + \frac{1}{b} \mathbb{E}\left[ (\phi_j^{i_{\max}})^2 \right] \\
= \quad & \frac{b-1}{bn(n-1)} \sum_{j_1, j_2 \in [n]} \phi_{j_1}^{i_{\max}} \phi_{j_2}^{i_{\max}} - \frac{b-1}{b(n-1)} \mathbb{E}\left[ (\phi_j^{i_{\max}})^2 \right] + \frac{1}{b} \mathbb{E}\left[ (\phi_j^{i_{\max}})^2 \right] \\
= \quad & \frac{b-1}{bn(n-1)} \sum_{j_1, j_2 \in [n]} \phi_{j_1}^{i_{\max}} \phi_{j_2}^{i_{\max}} - \beta(b) \mathbb{E}\left[ (\phi_j^{i_{\max}})^2 \right] \\
\overset{\text{①}}{=} \quad & \beta(b) \mathbb{E}\left[ (\phi_j^{i_{\max}})^2 \right] \\
\overset{\text{②}}{\leq} \quad & \beta(b) \mathbb{E}\left[ \|\phi_j\|_\infty^2 \right], \qquad \qquad \qquad \qquad \qquad \qquad \text{(H.6)}
\end{aligned}
$$

where ① is using the fact $\sum_{j=1}^{n} \phi_j^{i_{\max}} = 0$, ② is by definition of $\|\cdot\|_\infty$. Denote $i_{j_{\max}} = \arg\max_i |\phi_j^i|$. Hence

$$\mathbb{E}\left[\left\|\nabla f(x_{k+1}) - \tilde{\nabla}_{k+1}\right\|_\infty^2\right]$$

$$= \mathbb{E}\left[\left\|\frac{1}{b}\sum_{j\in\mathcal{B}}(\nabla f_j(x_{k+1}) - \nabla f_j(\tilde{x})) - (\nabla f(x_{k+1}) - \nabla f(\tilde{x}))\right\|_\infty^2\right]$$

$$\overset{①}{\leq} \beta(b)\mathbb{E}\left[\|\nabla f_j(x_{k+1}) - \nabla f_j(\tilde{x})) - (\nabla f(x_{k+1}) - \nabla f(\tilde{x})\|_\infty^2\right]$$

$$\overset{②}{=} \beta(b)\mathbb{E}\left[\left(\nabla_{i_{j_{\max}}} f_j(x_{k+1}) - \nabla_{i_{j_{\max}}} f_j(\tilde{x})) - \nabla_{i_{j_{\max}}} f(x_{k+1}) - \nabla_{i_{j_{\max}}} f(\tilde{x})\right)^2\right]$$

$$= \beta(b)\mathbb{E}\left[\left(\nabla_{i_{j_{\max}}} f_j(x_{k+1}) - \nabla_{i_{j_{\max}}} f_j(\tilde{x})\right)^2 - \left(\nabla_{i_{j_{\max}}} f(x_{k+1}) - \nabla_{i_{j_{\max}}} f(\tilde{x})\right)^2\right]$$

$$\leq \beta(b)\mathbb{E}\left[\left(\nabla_{i_{j_{\max}}} f_j(x_{k+1}) - \nabla_{i_{j_{\max}}} f_j(\tilde{x})\right)^2\right]$$

$$\overset{③}{\leq} \beta(b)\mathbb{E}\left[\|\nabla f_j(x_{k+1}) - \nabla f_j(\tilde{x})\|_\infty^2\right]$$

$$\leq 2\beta(b)L_1\mathbb{E}\left[f_j(\tilde{x}) - f_j(x_{k+1}) - \langle\nabla f_j(x_{k+1}), \tilde{x} - x_{k+1}\rangle\right]$$

$$\overset{④}{=} 2\beta(b)L_1(f(\tilde{x}) - f(x_{k+1}) - \langle\nabla f(x_{k+1}), \tilde{x} - x_{k+1}\rangle),$$

where ① is by (H.6), ② is using the fact $\mathbb{E}[(x - \mathbb{E}[x])^2] = \mathbb{E}[x^2] - (\mathbb{E}[x])^2$, ④ is by the definition of $\|\cdot\|_\infty$, ③ is by Lemma 15.

$\square$

**Lemma 19** (pCOMID). *Fixing $\tilde{\nabla}_{k+1}$ and letting*

$$(z_{k+1}, \theta_{k+1}) = pCOMID(\tilde{\nabla}_{k+1}, \theta_k, q, \lambda, \alpha), \tag{H.7}$$

*it satisfies for all $u \in \mathbb{R}^d$,*

$$\alpha\langle\tilde{\nabla}_{k+1}, z_{k+1} - u\rangle + \alpha\lambda\|z_{k+1}\|_1 - \alpha\lambda\|u\|_1 \leq -B_p(z_{k+1}, z_k) + B_p(u, z_k) - B_p(u, z_{k+1}). \tag{H.8}$$

*Proof.* From [13], we known that pCOMID exactly solves the following mirror descent problem,

$$z_{k+1} = \arg\min_z\{\langle\tilde{\nabla}_{k+1}, z - z_k\rangle + \frac{1}{\alpha}B_p(z, z_k) + \lambda\|z\|_1\}. \tag{H.9}$$

By the optimality condition of $z_{k+1}$, it follows that

$$\nabla\frac{1}{2}\|z_{k+1}\|_p^2 - \nabla\frac{1}{2}\|z_k\|_p^2 + \alpha\tilde{\nabla}_{k+1} + \alpha g = 0,$$

where $g \in \partial\lambda\|z_{k+1}\|_1$. Then the equality

$$\langle\nabla\frac{1}{2}\|z_{k+1}\|_p^2 - \nabla\frac{1}{2}\|z_k\|_p^2 + \alpha\tilde{\nabla}_{k+1} + \alpha g, z_{k+1} - u\rangle = 0$$

holds. In addition by Lemma 16, it follows that $\langle\nabla\frac{1}{2}\|z_{k+1}\|_p^2 - \nabla\frac{1}{2}\|z_k\|_p^2, z_{k+1} - u\rangle = B_p(z_{k+1}, z_k) - B_p(u, z_k) + B_p(u, z_{k+1})$. By the convexity of $\lambda\|z\|_1$, $\langle g, z_{k+1} - u\rangle \geq \lambda\|z_{k+1}\|_1 - \lambda\|u\|_1$. Therefore, we can write

$$\alpha\langle\tilde{\nabla}_{k+1}, z_{k+1} - u\rangle + \alpha\lambda\|z_{k+1}\|_1 - \alpha\lambda\|u\|_1$$

$$= -\langle\nabla\frac{1}{2}\|z_{k+1}\|_p^2 - \nabla\frac{1}{2}\|z_k\|_p^2, z_{k+1} - u\rangle - \langle\alpha g, z_{k+1} - u\rangle + \alpha\lambda\|z_{k+1}\|_1 - \alpha\lambda\|u\|_1$$

$$\leq -B_p(z_{k+1}, z_k) + B_p(u, z_k) - B_p(u, z_{k+1}).$$

$\square$

**Lemma 20** (Couping step 1). *If* $x_{k+1} = \tau_1 z_k + \tau_2 \tilde{x} + (1 - \tau_1 - \tau_1) y_k$, *where* $\tau_1 \leq \frac{1}{(1+2\beta(b))\alpha L_1}$ *and* $\tau_2 = \frac{1}{2}$,

$$\alpha \langle \nabla f(x_{k+1}), z_k - u \rangle - \alpha\lambda\|u\|_1$$

$$\leq \quad \frac{\alpha}{\tau_1}\left(F(x_{k+1}) - \mathbb{E}[F(y_{k+1})] + \tau_2 F(\tilde{x}) - \tau_2 \mathbb{E}[F(x_{k+1})] - \tau_2 \langle \nabla f(x_{k+1}), \tilde{x} - x_{k+1}\rangle\right)$$

$$+ B_p(u, z_k) - \mathbb{E}[B_p(u, z_{k+1})] + \frac{\alpha(1 - \tau_1 - \tau_2)}{\tau_1}\lambda\|y_k\|_1 - \frac{\alpha}{\tau_1}\lambda\|x_{k+1}\|_1.$$

*Proof.* It follows taht

$$\alpha \langle \tilde{\nabla}_{k+1}, z_k - u \rangle + \alpha\lambda\|z_{k+1}\|_1 - \alpha\lambda\|u\|_1$$

$$= \quad \alpha \langle \tilde{\nabla}_{k+1}, z_k - z_{k+1} \rangle + \alpha \langle \tilde{\nabla}_{k+1}, z_{k+1} - u \rangle + \alpha\lambda\|z_{k+1}\|_1 - \alpha\lambda\|u\|_1$$

$$\overset{①}{\leq} \quad \alpha \langle \tilde{\nabla}_{k+1}, z_k - z_{k+1} \rangle - B_p(z_{k+1}, z_k) + B_p(u, z_k) - B_p(u, z_{k+1})$$

$$\overset{②}{\leq} \quad \alpha \langle \tilde{\nabla}_{k+1}, z_k - z_{k+1} \rangle - \frac{p-1}{2}\|z_{k+1} - z_k\|_p^2 + B_p(u, z_k) - B_p(u, z_{k+1})$$

$$\overset{③}{\leq} \quad \alpha \langle \tilde{\nabla}_{k+1}, z_k - z_{k+1} \rangle - \frac{p-1}{2}d^{-(1-\frac{1}{p})}\|z_{k+1} - z_k\|_1^2 + B_p(u, z_k) - B_p(u, z_{k+1}),$$

$$\overset{④}{=} \quad \alpha \langle \tilde{\nabla}_{k+1}, z_k - z_{k+1} \rangle - \frac{1}{2C}\|z_{k+1} - z_k\|_1^2 + B_p(u, z_k) - B_p(u, z_{k+1}), \qquad \text{(H.10)}$$

where ① is by Lemma 19, ② is by Lemma 16, ③ is by Lemma 14 and ④ is by the setting $C = \frac{d^{\frac{2\delta}{1+\delta}}}{\delta}$ and $p = 1 + \delta$ in Alg. 2.

By defining $v \overset{def}{=} \tau_1 z_{k+1} + \tau_2 \tilde{x} + (1 - \tau_1 - \tau_2) y_k$, we have $x_{k+1} - v = \tau_1(z_k - z_{k+1})$ and therefore

$$\mathbb{E}[\alpha \langle \tilde{\nabla}_{k+1}, z_k - z_{k+1} \rangle - \frac{1}{2C}\|z_{k+1} - z_k\|_1^2] = \mathbb{E}[\frac{\alpha}{\tau_1}\langle \tilde{\nabla}_{k+1}, x_{k+1} - v \rangle - \frac{1}{2C\tau_1^2}\|x_{k+1} - v\|_1^2]$$

$$= \quad \mathbb{E}\left[\frac{\alpha}{\tau_1}\left(\langle \tilde{\nabla}_{k+1}, x_{k+1} - v \rangle - \frac{1}{2C\alpha\tau_1}\|x_{k+1} - v\|_1^2 - \lambda\|v\|_1 + \lambda\|x_{k+1}\|_1\right)\right.$$

$$\left. + \frac{\alpha}{\tau_1}\left(\lambda\|v\|_1 - \lambda\|x_{k+1}\|_1\right)\right]$$

$$\overset{①}{=} \quad \mathbb{E}\left[\frac{\alpha}{\tau_1}\left(\langle \tilde{\nabla}_{k+1}, x_{k+1} - v \rangle - \frac{(1+2\beta(b))L_1}{2}\|x_{k+1} - v\|_1^2 - \lambda\|v\|_1 + \lambda\|x_{k+1}\|_1\right)\right.$$

$$\left. + \frac{\alpha}{\tau_1}\left(\lambda\|v\|_1 - \lambda\|x_{k+1}\|_1\right)\right]$$

$$\overset{②}{\leq} \quad \mathbb{E}\left[\frac{\alpha}{\tau_1}\left(F(x_{k+1}) - F(y_{k+1}) + \frac{1}{4\beta(b)L_1}\sigma_{k+1}^2\right) + \frac{\alpha}{\tau_1}\left(\lambda\|v\|_1 - \lambda\|x_{k+1}\|_1\right)\right]$$

$$\overset{③}{\leq} \quad \mathbb{E}\left[\frac{\alpha}{\tau_1}\left(F(x_{k+1}) - F(y_{k+1}) + \frac{1}{2}(f(\tilde{x}) - f(x_{k+1}) - \langle \nabla f(x_{k+1}, \tilde{x} - x_{k+1}\rangle)\right)\right.$$

$$\left. + \frac{\alpha}{\tau_1}(\tau_1\lambda\|z_{k+1}\|_1 + \tau_2\lambda\|\tilde{x}\|_1 + (1 - \tau_1 - \tau_2)\lambda\|y_k\|_1 - \lambda\|x_{k+1}\|_1)\right], \qquad \text{(H.11)}$$

where ① is by the setting $\alpha\tau_1 = \frac{1}{(1+2\beta(b))CL_1}$, ② is by Lemma 17, ③ is by Lemma 18 and the convexity $\|v\|_1 = \|\tau_1 z_{k+1} + \tau_2 \tilde{x} + (1 - \tau_1 - \tau_2)y_k\|_1 \leq \tau_1\|z_{k+1}\|_1 + \tau_2\|\tilde{x}\|_1 + (1 - \tau_1 - \tau_2)\|y_k\|_1$. Then, it is showed that $\mathbb{E}[\langle \tilde{\nabla}_{k+1}, z_k - u \rangle] = \langle \nabla f(x_{k+1}, z_k - u\rangle$ and $\tau_2 = \frac{1}{2}$. Combing (H.10) and (H.11), Lemma 20 is obtained. $\qquad \square$

**Lemma 21** (Coupling step 2). *Under the same choices of* $\tau_1, \tau_2$ *as in Lemma 20, one has*

$$0 \leq \frac{\alpha(1 - \tau_1 - \tau_2)}{\tau_1}(F(y_k) - F(x^*)) - \frac{\alpha}{\tau_1}(\mathbb{E}[F(y_{k+1})] - F(x^*)) + \frac{\alpha\tau_2}{\tau_1}(F(\tilde{x}) - F(x^*))$$

$$+ B_p(x^*, z_k) - \mathbb{E}[B_p(x^*, z_{k+1})].$$

*Proof.* It follows that

$$\alpha(f(x_{k+1}) - f(u)) \overset{①}{\leq} \alpha\langle \nabla f(x_{k+1}), x_{k+1} - u\rangle$$

$$= \alpha\langle \nabla f(x_{k+1}), x_{k+1} - z_k\rangle + \alpha\langle \nabla f(x_{k+1}), z_k - u\rangle$$

$$\overset{②}{=} \frac{\alpha\tau_2}{\tau_1}\langle \nabla f(x_{k+1}), \tilde{x} - x_{k+1}\rangle + \frac{\alpha(1 - \tau_1 - \tau_2)}{\tau_1}\langle \nabla f(x_{k+1}), y_k - x_{k+1}\rangle + \alpha\langle \nabla f(x_{k+1}), z_k - u\rangle$$

$$\overset{③}{\leq} \frac{\alpha\tau_2}{\tau_1}\langle \nabla f(x_{k+1}), \tilde{x} - x_{k+1}\rangle + \frac{\alpha(1 - \tau_1 - \tau_2)}{\tau_1}(f(y_k) - f(x_{k+1})) + \alpha\langle f(x_{k+1}), z_k - u\rangle, \text{(H.12)}$$

where ① is by the convexity of $f(x)$, ② is by the convex combination $x_{k+1} = \tau_1 z_k + \tau_2\tilde{x} + (1 - \tau_1 - \tau_2)y_k$, ③ is again by the convexity of $f(x)$. Applying Lemma 20 to (H.12), it follows that

$$\alpha(f(x_{k+1}) - F(u)) \leq \frac{\alpha(1 - \tau_1 - \tau_2)}{\tau_1}(F(y_k) - f(x_{k+1}))$$

$$+ \frac{\alpha}{\tau_1}(F(x_{k+1}) - \mathbb{E}[F(y_{k+1})] + \tau_2 F(\tilde{x}) - \tau_2 f(x_{k+1})) + B_p(u, z_k) - \mathbb{E}[B_p(u, z_{k+1})] - \frac{\alpha}{\tau_1}\lambda\|x_{k+1}\|_1,$$

which implies

$$\alpha(F(x_{k+1}) - F(u)) \leq \frac{\alpha(1 - \tau_1 - \tau_2)}{\tau_1}(F(y_k) - F(x_{k+1}))$$

$$+ \frac{\alpha}{\tau_1}(F(x_{k+1}) - \mathbb{E}[F(y_{k+1})] + \tau_2 F(\tilde{x}) - \tau_2 F(x_{k+1})) + B_p(u, z_k) - \mathbb{E}[B_p(u, z_{k+1})].$$

After arrangement and setting $u$ to some minimizer $x^*$, Lemma 21 is obtained. $\qquad\square$

### H.2.2    Proof of Theorem 2

*Proof.* Assume the parameter $\tau_{1,s}$ and $\alpha_s$ satisfies the assumption $\tau_{1,s}\alpha_s = \frac{1}{(1+2\beta(b))CL_1}$ in Lemma 20. Let $D_k \overset{def}{=} F(y_k) - F(x^*)$ and $\tilde{D}^s \overset{def}{=} F(\tilde{x}^s) - F(x^*)$, Lemma 21 can be rewritten as

$$0 \leq \frac{\alpha_s(1 - \tau_{1,s} - \tau_2)}{\tau_{1,s}}D_k - \frac{\alpha_s}{\tau_{1,s}}\mathbb{E}[D_{k+1}] + \frac{\alpha_s\tau_2}{\tau_{1,s}}\tilde{D}^s + B_p(x^*, z_k) - \mathbb{E}[B_p(x^*, z_{k+1})].$$

In the $s$-th epoch, summing up the above inequality for all the iterations $k = sm, sm + 1, \ldots, sm + m - 1$, it follows that

$$\mathbb{E}\left[\alpha_s\frac{1 - \tau_{1,s} - \tau_2}{\tau_{1,s}}D_{(s+1)m} + \alpha_s\frac{\tau_{1,s} + \tau_2}{\tau_{1,s}}\sum_{l=1}^{m}D_{sm+l}\right]$$

$$\leq \quad \alpha_s\frac{1 - \tau_{1,s} - \tau_2}{\tau_{1,s}}D_{sm} + \alpha_s\frac{\tau_2}{\tau_{1,s}}m\tilde{D}^s + B_p(x^*, z_{sm}) - \mathbb{E}[B_p(x^*, z_{(s+1)m})]. \text{ (H.13)}$$

It should be noted that in (H.13), we fix all the randomness in the first $s - 1$ epochs and take expectation on the current epoch $s$.

By the definition $\tilde{x}^s = \frac{1}{m}\sum_{l=1}^{m}y_{(s-1)m+l}$ in Alg. 2 and the convexity of $F(x)$, we have $m\tilde{D}^s \leq \sum_{l=1}^{m}D_{(s-1)m+l}$. Then for each $s \geq 1$, by (H.13), it follows that

$$\mathbb{E}\left[\frac{1}{\tau_{1,s}^2}D_{(s+1)m} + \frac{\tau_{1,s} + \tau_2}{\tau_{1,s}^2}\sum_{l=1}^{m-1}D_{sm+l}\right]$$

$$\leq \quad \frac{1 - \tau_{1,s}}{\tau_{1,s}^2}D_{sm} + \frac{\tau_2}{\tau_{1,s}^2}\sum_{l=1}^{m-1}D_{(s-1)m+l} + (1 + 2\beta(b))CLB_p(x^*, z_{sm}) - (1 + 2\beta(b))CL\mathbb{E}[B_p(x^*, z_{(s+1)m})].$$

For $s = 0$, (H.13) can be written as

$$\mathbb{E}\left[\frac{1}{\tau_{1,0}^2}D_m + \frac{\tau_{1,0} + \tau_2}{\tau_{1,0}^2}\sum_{l=1}^{m-1}D_l\right]$$

$$\leq \quad \frac{1 - \tau_{1,0} - \tau_2}{\tau_{1,0}^2}D_0 + \frac{\tau_2 m}{\tau_{1,0}^2}\tilde{D}^0 + (1 + 2\beta(b))CL_1 B_p(x^*, z_0)$$

$$\quad - (1 + 2\beta(b))CL_1\mathbb{E}[B_p(x^*, z_m)]. \tag{H.14}$$

Choose $\tau_{1,s} = \frac{2}{s+4} \leq \frac{1}{2}$ which satisfies

$$\frac{1}{\tau_{1,s}^2} \geq \frac{1 - \tau_{1,s+1}}{\tau_{1,s+1}^2} \quad \text{and} \quad \frac{\tau_{1,s} + \tau_2}{\tau_{1,s}^2} \geq \frac{\tau_2}{\tau_{1,s+1}^2}. \tag{H.15}$$

Then it follows that

$$\mathbb{E}\left[ \frac{m}{\tau_{1,S-1}^2} \tilde{D}^S + (1 + 2\beta(b)) \, CL_1 B_p(x^*, z_{Sm}) \right]$$

$$\overset{\text{①}}{\leq} \mathbb{E}\left[ \frac{1}{\tau_{1,S-1}^2} D_{Sm} + \frac{\tau_{1,S-1}}{\tau_{1,S-1}^2} \sum_{l=1}^{m-1} D_{(S-1)m+l} + (1 + 2\beta(b)) \, CL_1 B_p(x^*, z_{Sm}) \right]$$

$$\overset{\text{②}}{\leq} \mathbb{E}\left[ \frac{1}{\tau_{1,S-1}^2} D_{Sm} + \frac{\tau_2}{\tau_{1,S}^2} \sum_{l=1}^{m-1} D_{(S-1)m+l} + (1 + 2\beta(b)) \, CL_1 B_p(x^*, z_{Sm}) \right]$$

$$\overset{\text{③}}{\leq} \frac{1 - \tau_{1,0} - \tau_2}{\tau_{1,0}^2} D_0 + \frac{\tau_2 m}{\tau_{1,0}^2} \tilde{D}^0 + (1 + 2\beta(b)) \, CL_1 B_p(x^*, z_0)$$

$$\overset{\text{④}}{=} \frac{\tau_2 m}{\tau_{1,0}^2} \tilde{D}^0 + (1 + 2\beta(b)) \, CL_1 B_p(x^*, z_0),$$

where ① is by $m\tilde{D}^s \leq \sum_{l=1}^m D_{(s-1)m+l}$, ② is by $\tau_2 \geq \tau_{1,S-1} \geq \tau_{1,S}$, ③ uses (H.15) to telescope (H.13) and (H.14) for all $s = 0, 1, \ldots, S-1$ and ④ is by $\tau_{1,0} = \tau_2 = \frac{1}{2}$.

$$\mathbb{E}[F(\tilde{x}^S) - F(x^*)]$$

$$= \mathbb{E}[\tilde{D}^S] \leq \frac{\tau_{1,S-1}^2}{m} \cdot \left( \frac{\tau_2 m}{\tau_{1,0}^2} \tilde{D}^0 + (1 + 2\beta(b)) \, CL_1 B_p(x^*, z_0) \right)$$

$$= \frac{4}{m(S+3)^2} \left( 2m(F(\tilde{x}^0) - F(x^*)) + (1 + 2\beta(b)) \, CL_1 B_p(x^*, z_0) \right). \tag{H.16}$$

Meanwhile, by setting $\tilde{x}_0 = z_0 = 0$, using the optimality condition $0 \in \nabla f(x^*) + \partial \lambda \|x^*\|_1$ we have

$$F(\tilde{x}^0) - F(x^*) \quad = \quad f(0) - f(x^*) - \lambda \|x^*\|_1$$

$$\overset{\text{①}}{\leq} \quad \langle \nabla f(x^*), 0 - x^* \rangle + \frac{L_1}{2} \|x^*\|_1^2 - \lambda \|x^*\|_1$$

$$\overset{\text{②}}{=} \quad \langle -\partial \lambda \|x^*\|_1, -x^* \rangle + \frac{L_1}{2} \|x^*\|_1^2 - \lambda \|x^*\|_1$$

$$\overset{\text{③}}{\leq} \quad \|\partial \lambda \|x^*\|_1 \|_\infty \|x^*\|_1 + \frac{L_1}{2} \|x^*\|_1^2 - \lambda \|x^*\|_1$$

$$\overset{\text{④}}{\leq} \quad \lambda \|x^*\|_1 + \frac{L_1}{2} \|x^*\|_1^2 - \lambda \|x^*\|_1$$

$$= \quad \frac{L_1}{2} \|x^*\|_1^2, \tag{H.17}$$

where ① is by the smoothness assumption of $f(x)$, ② is by selecting the subgradient of $\lambda \|x^*\|_1$ with $-\partial \lambda \|x^*\|_1 = \nabla f(x^*)$, ③ is by lemma 13, ④ is by using the property of subgradient $\|\partial \lambda \|x^*\|_1 \|_\infty \leq \lambda$. In addition, for $1 < p \leq 2$,

$$B_p(x^*, z_0) \quad = \quad B_p(x^*, 0) = \frac{1}{2} \|x^*\|_p^2 - \frac{1}{2} \|0\|_p^2 - \langle \nabla \frac{1}{2} \|0\|_p^2, x^* - 0 \rangle$$

$$= \quad \frac{1}{2} \|x^*\|_p^2 \leq \frac{1}{2} \|x^*\|_1^2. \tag{H.18}$$

Furthermore, minimizing $C = \frac{d^{\frac{2\delta}{1+\delta}}}{\delta}$ w.r.t $\delta$, we get $\delta = \log(d) - 1 - \sqrt{(\log(d) - 1)^2 - 1}$ and $p = 1 + \delta = \log(d) - \sqrt{(\log(d) - 1)^2 - 1} \in (1, 2]$,

Then combing (H.16), (H.17) and (H.18), we get the final result.

$$\mathbb{E}[F(\tilde{x}^S) - F(x^*)] \quad \leq \quad \frac{4}{(S+3)^2} \left( 1 + \frac{1 + 2\beta(b)}{2m} C \right) L_1 \|x\|_1^2. \tag{H.19}$$

$\square$