[Reviews · NeurIPS 2017]

Reviewer 1



Paper Summary: The main idea is that Nesterov's acceleration method's and Stochastic Gradient Descent's (SGD) advantages are used to solve sparse and dense optimization problems with high-dimensions by using an improved GCD (Greedy Coordinate Descent) algorithm. First, by using a greedy rule, an $l_1$-square-regularized approximate optimization problem (find a solution close to $x^*$ within a neighborhood $\epsilon$) can be reformulated as a convex but non-trivial to solve problem. Then, the same problem is solved as an exact problem by using the SOTOPO algorithm. Finally, the solution is improved by using both the convergence rate advantage of Nesterov's method and the "reduced-by-one-sample" complexity of SGD. The resulted algorithm is an improved GCD (ASGCD=Accelerated Stochastic Greedy Coordinate Descent) with a convergence rate of $O(\sqrt{1/\epsilon})$ and complexity reduced-by-one-sample compared to the vanilla GCD. Originality of the paper: The SOTOPO algorithm proposed, takes advantage of the l1 regularization term to investigate the potential values of the sub-gradient directions and sorts them to find the optimal direction without having to calculate the full gradient beforehand. The combination of Nesterov's advantage with SGC advantage and the GCD advantage is less impressive. Bonus for making an efficient and rigorous algorithm despite the many pieces that had to be put together Contribution: -Reduces complexity and increases convergence rate for large-scale, dense, convex optimization problems with sparse solutions (+), -Uses existing results known to improve performance and combines them to generate a more efficient algorithm (+), -Proposes a criterion to reduce the complexity by identifying the non-zero directions of descent and sorting them to find the optimal direction faster (+), -Full computation of the gradient beforehand is not necessary in the proposed algorithm (+), -There is no theoretical way proposed for the choice of the regularization parameter $\lambda$ as a function of the batch size. The choice of $\lambda$ seems to affect the performance of the ASGCD in both batch choice cases (-). Technical Soundness: -All proofs to Lemmas, Corollaries, Theorems and Propositions used are provided in the supplementary material (+), -Derivations are rigorous enough and solid. In some derivations further reference to basic optimization theorems or Lemmas could be more en-lighting to non-optimization related researchers (-). Implementation of Idea: The algorithm is complicated to implement (especially the SOTOPO part). Clarity of presentation: -Overall presentation of the paper is detailed but the reader is not helped to keep in mind the bigger picture (might be lost in the details). Perhaps reminders of the goal/purpose of each step throughout the paper would help the reader understand why each step is necessary(-), -Regarding the order of application of different known algorithms or parts of them to the problem: it is explained but could be more clear with a diagram or pseudo-code (-), -Notation: in equation 3, $g$ is not clearly explained and in Algorithm 1 there are two typos in referencing equations (-), -For the difficulty of writing such a mathematically incremental paper, the clarity is at descent (+). Theoretical basis: -All Lemmas and transformations are proved thoroughly in the supplementary material (+), -Some literature results related to convergence rate or complexity of known algorithms are not referenced (lines 24,25,60,143 and 73 was not explained until equation 16 which brings some confusion initially). Remark 1 could have been referenced/justified so that it does not look completely arbitrary (-), -A comparison of the theoretical solution accuracy with the other pre-existing methods would be interesting to the readers (-), -In the supplementary material in line 344, a $d \theta_t$ is missing from one of the integrals (-). Empirical/Experimental basis: -The experimental results verify the performance of the proposed algorithm with respect to the ones chosen for comparison. Consistency in the data sets used between the different algorithms, supports a valid experimental analysis (+), -A choice of better smoothing constant $T_1$ is provided in line 208 (+) but please make it more clear to the reader why this is a better option in the case of $b=n$ batch size (-), -The proposed method is under-performing (when the batch size is 1) compared to Katyusha for small regularization $10^{-6}$ and for the test case Mnist while for Gisette it is comparable to Katyusha. There might be room for improvement in these cases or if not it would be interesting to show which regularization value is the threshold and why. The latter means that the algorithm proposed is more efficient for large-scale problems with potentially a threshold in sparsity (minimum regularization parameter) that the authors have not theoretically explored. Moreover, there seems to be a connection between the batch size (1 or n, in other words stochastic or deterministic case) and the choice of regularization value that makes the ASGCD outperform other methods which is not discussed (-). Interest to NIPS audience [YES]: This paper compares the proposed algorithm with well-established algorithms or performance improvement schemes and therefore would be interesting to the NIPS audience. Interesting discussion might arise related to whether or not the algorithm can be simplified without compromising it's performance.

Reviewer 2



This paper aims to solve L1 regularized ERM problem. The developments in this paper seem to be motivated by the desire to combine several successful techniques into a single algorithm: greedy coordinate descent, SVRG and acceleration. Accelerated SVRG is known as Katyusha and hence the main task is to combine Katyusha (which would randomize over a minibatch of examples n) with greedy coordinate descent (which would update a subset of the d feature vectors in a “greedy” fashion). The way this is done in this paper is interesting as the solution is surprisingly simple and effective. The authors observe (by citing older literature) that without L1 regularization, and in the batch setting, greedy coordinate descent can be expressed as a gradient descent step if instead of the standard L2 norm in the upper bound one used the L1 norm (one also needs to change the scaling/Lipschitz constant). By adding the L1 regularizer, this property is lost, and a subset of variables might be updated. However, the resulting method could still be interpreted as a variant of greedy coordinate descent. This strategy is then combined with Katyusha and the result is an accelerated (via Katyusha momentum), stochastic (over n) and greedy (over d) method. The authors show that in some settings, the resulting complexity can beat that of Katyusha itself. It is important the the authors are able to devise a fast method for solving the subproblems involved. The key subproblem, (3), is solved via a novel, nontrivial and efficient method called SOTOPO. The starting point here is a variational reformulation of the squared L1 norm as a convex minimization problem over the unit simplex. The authors then write the problem as a min-min problem in the original and auxiliary variables. Switching the order of taking the min ultimately leads to an efficient method for solving (3). This seems of independent interest, which is good. I like the paper. It is well written. It presents some interesting novel ideas, leads to an efficient method, works in practice in several regimes (interestingly, for both n > > d and n < < d regimes, although this should be investigated in more detail), and also leads to improved complexity for the L1 regularized ERM problem in certain regimes. Some comments: 1) Line 31: There are some earlier contributions to accelerated randomized coordinate descent than [11, 20]. The first is due to tNesterov [2012], but suffered from expensive iterations. This was remedied by Lee and Sidford [arXiv: 1305.1922] and Fercoq & Richtarik [arXiv: 1312.5799]. Further improvements were made with the introduction of nonuniform probabilities into accelerated randomized coordinate descent. This was done by Qu and Richtarik [arXiv: 1412.8060] and then extended to strongly convex setting by Allen-Zhu, Qu, Richtarik and Yuan [arXiv: 1512.09103], and later independently by Nesterov and Stich [2016]. 2) The setup of this paper reminds me of a similar synthesis of two methods: SVRG and randomized coordinate descent. This was done in Konecny, Qu & Richtarik [arXiv:1412.6293]. The difference here is that their method is not accelerated (Katyusha did not exist then), and instead of greedy, they use randomized coordinate descent. I am wondering what the connections are. 3) Regarding experiments: the APPROX method of Fercoq and Richtarik [arXiv:1312.5799] should be included. This is a batch method in n, but capable to randomize over subsets of the d variables, and capable of utilizing sparsity in the data. This method should do very well in the high d and low n regime. 4) Explain the effect of \eta on the sparsity level in subproblem (3) – even with lambda = 0. Clearly, if eta is small enough, the bound used upper bounds the standard quadratic approximation. In such a case, SOTOPO is not needed; and Katyusha applies directly. There is a thin line here: it will be useful to comment on L1 vs L2 and so on in light of this. Some minor comments: 1) What are (???) in Step 1 of Alg 1? 2) 19 and 20: are referred to -> refer to 3) 19: that uses some algebra trick to -> using an elusive algebra trick to 4) 20: samples -> sample 5) 27: resulted -> resulting {this appears in many places in the paper, such as lines 51, 56, …} 6) 29: reduces -> reduce 7) 38: preferable than -> preferable to 8) 40: GSD has much -> GCD to have much 9) 50: entries -> entry 10) 55: sqoximation -> approximation 11) 88: is -> are 12) 99: While -> Since 13) 121: of the -> of === post rebuttal feedback === I am keeping my decision.

Reviewer 3



This paper proposes a new greedy coordinate descent type algorithm. The algorithm uses a novel Gauss-Southwell coordinate selection rule, where a convex minimization problem involving the l_1-norm squared is used to select the coordinates to be updated. This coordinate selection strategy is interesting, and it has been developed with practicality in mind, which is important. The authors have also incorporated an acceleration strategy, and presented the ASGCD algorithm (Accelerated Stochastic Greedy Coordinate Descent). Greedy coordinate selection schemes are very popular at the moment, and I think there will be many authors interested in this strategy. However, there seem to be quite a few typos in the paper, and the standard of English should be improved. I suggest that the authors thoroughly proofread the papers to correct these errors. Also, the numerical experiments seem to support the proposed algorithm but the problems being tested are quite small, and it would have been good for the authors to have evaluated the algorithm on some large-scale problems.